# Deep learning-based transformation of H&E stained tissues into special stains

Kevin de Haan [1,2,3], Yijie Zhang[1,2,3], Jonathan E. Zuckerman[4], Tairan Liu[1,2,3], Anthony E. Sisk[4], Miguel F. P. Diaz[5], Kuang-Yu Jen [6], Alexander Nobori[4], Sofia Liou[4], Sarah Zhang [4], Rana Riahi[4], Yair Rivenson [1,2,3 ✉], W. Dean Wallace [7 ✉] & Aydogan Ozcan [1,2,3,8 ✉]

Pathology is practiced by visual inspection of histochemically stained tissue slides. While the hematoxylin and eosin (H&E) stain is most commonly used, special stains can provide additional contrast to different tissue components. Here, we demonstrate the utility of supervised learning-based computational stain transformation from H&E to special stains (Masson's Trichrome, periodic acid-Schiff and Jones silver stain) using kidney needle core biopsy tissue sections. Based on the evaluation by three renal pathologists, followed by adjudication by a fourth pathologist, we show that the generation of virtual special stains from existing H&E images improves the diagnosis of several non-neoplastic kidney diseases, sampled from 58 unique subjects (P = 0.0095). A second study found that the quality of the computationally generated special stains was statistically equivalent to those which were histochemically stained. This stain-to-stain transformation framework can improve preliminary diagnoses when additional special stains are needed, also providing significant savings in time and cost.

[1] Electrical and Computer Engineering Department, University of California, Los Angeles, CA, USA. [2] Bioengineering Department, University of California, Los Angeles, CA, USA. [3] California NanoSystems Institute (CNSI), University of California, Los Angeles, CA, USA. [4] Department of Pathology and Laboratory Medicine, David Geffen School of Medicine, University of California, Los Angeles, Los Angeles, CA, USA. [5] Kaiser Permanente Los Angeles Medical Center, Department of Pathology, Los Angeles, CA, USA. [6] Department of Pathology and Laboratory Medicine, University of California at Davis, Sacramento, CA, USA. [7] Department of Pathology and Laboratory Medicine, Keck School of Medicine of USC, Los Angeles, CA, USA. [8] Department of Surgery, David Geffen School of Medicine, University of California, Los Angeles, CA, USA. ✉email: rivensonyair@g.ucla.edu; william.wallace@med.usc.edu; ozcan@ucla.edu

Histological analysis of stained human tissue samples is the gold standard for evaluation of many diseases, as the fundamental basis of any pathologic evaluation is the examination of histologically stained tissue affixed on a glass slide using either a microscope or a digitized version of the histologic image following the image capture by a whole slide image (WSI) scanner. The histological staining step is a critical part of the pathology workflow and is required to provide contrast and color to tissue by facilitating a chromatic distinction among different tissue constituents. The most common stain (otherwise referred to as the routine stain) is the hematoxylin and eosin (H&E), which is applied to nearly all clinical cases, covering ~80% of all the human tissue staining performed globally[1]. The H&E stain is relatively easy to perform and is widely used across the industry. In addition to H&E, there are a variety of other histological stains with different properties which are used by pathologists to better highlight different tissue constituents. For example, Masson's trichrome (MT) stain is used to view connective tissue[2] and periodic acid-Schiff (PAS) can be used to better scrutinize basement membranes. The black staining in the Jones methenamine silver (JMS) stain offers a sharp contrast to visualize glomerular architecture and enables the pathologist to recognize subtle basement membrane abnormalities resulting from remodeling due to various forms of injury. These features have importance for certain disease types such as nonneoplastic kidney disease[3]. These non-H&E stains are also called special stains and their use is the standard of care in the pathologic evaluation of certain disease entities including nonneoplastic kidney, liver, and lung diseases, among others.

The traditional histopathology workflow can be time-consuming, expensive, and requires laboratory infrastructure. Tissue must first be sampled from the patient, fixed either through freezing in optimal cutting temperature (OCT) compound, or paraffin embedding, sliced into thin (2–10 μm) sections, and mounted onto a glass slide. Only then can these sections be stained using the desired chemical staining procedure. Furthermore, if multiple stains are needed, multiple tissue sections are cut, and a separate procedure must be used for each stain. While H&E staining is performed using a streamlined staining procedure, the special stains often require more preparation time, effort, and monitoring by a histotechnologist, which increases the cost of the procedure and takes additional time to produce. This can in turn increase the time for diagnosis, especially when a pathologist determines that these additional special stains are needed after the H&E stained tissue has been examined. The tissue sectioning and staining procedure may therefore need to be repeated for each special stain, which is wasteful in terms of resources, materials, and might place a burden on both the healthcare system and patients if there is an urgent need for a diagnosis.

Recognizing some of these limitations, different approaches have been developed to improve the histopathology workflow. Histological staining has been reproduced by imaging rapidly labeled tissue sections (usually by a nuclear staining dye) using an alternative contrast mechanism acquired by e.g., nonlinear microscopy[4] or ultraviolet tissue surface excitation[5], and digitally transforming the captured images into user-calibrated H&E-like images[6]. These approaches mainly focus on eliminating tissue fixation from the workflow, targeting rapid intraoperative contrast to unfixed specimens. More recently, computational staining techniques known as virtual staining have been developed. Using deep learning, virtual staining has been applied on label-free (i.e., unstained) fixed and glass slide affixed tissue sections using various modalities such as autofluorescence[7,8], hyperspectral imaging[9], quantitative phase imaging[10], and others[11,12]. Virtual staining of label-free tissue not only has the ability to reduce costs and allow for faster staining, but also allows the user to perform further advanced analysis on the tissue since the destructive additional sectioning and staining process is avoided that can cause the specimen to be depleted leading to e.g., additional/unnecessary biopsies from the patients[13]. Furthermore, virtual staining of label-free tissue enables new capabilities such as the use of multiple virtual stains upon a single tissue section, stain normalization (i.e., standardization), the region-of-interest specific digital blending of multiple stains, all of which are challenging or highly impractically with standard histochemical staining workflows[7,8].

An alternative approach that can be used to bypass histochemical tissue staining is to computationally transform the WSI of an already stained tissue into another stain (this will be referred to as stain transformation). This allows users to reduce the number of physical stains required without making any changes to their traditional histopathology workflow, and also carries many of the benefits of the virtual staining techniques such as improving stain consistency and reduction in stain preparation time. Different stain transformations have been demonstrated in the literature, e.g., the transformation of H&E into MT[14] or transformation of fibroblast activation protein-cytokeratin (FAP-CK), duplex immunohistochemistry (IHC) protocol[15], from images of Ki67-CD8 stained slides. Stain transformations have also been used as a tool to improve the effectiveness of image segmentation algorithms[16,17]. However, many of these stain transformation techniques rely upon unsupervised approaches which use distribution matching losses used by techniques such as cycle consistent generative adversarial networks (GANs)—also known as CycleGANs[18]. It has been shown that, when applied to medical imaging, neural networks trained using only these types of distribution matching losses are prone to hallucinations[19]. Some researchers have been able to avoid the use of these distribution matching losses and unpaired image data by training networks to perform other stain-to-stain transformations. For example, a stain transformation network was trained using image pairs acquired from adjacent tissue sections[20], while another work used image pairs captured by chemically destaining and then restaining the same tissue sections[21].

In this paper, we present a supervised deep learning-based stain transformation framework, outlined in Fig. 1. The training of this technique is based on spatially-registered (i.e., perfectly paired) image datasets which allow the stain transformation network to be trained without relying on unpaired image data and corresponding distribution matching losses. We demonstrate the efficacy of this technique by evaluating kidney tissues with various nonneoplastic diseases. Nonneoplastic kidney disease relies on special stains to provide the standard of care pathologic evaluation. In many clinical practices, H&E stains are available well before the special stains are prepared, and pathologists may provide a preliminary diagnosis to enable the patient's nephrologist to begin any necessary treatment. In a setting when only H&E slides are initially available, the preliminary diagnosis is followed by the final diagnosis made by examining the special stain images, which are often provided the next working day. Using the presented stain transformation technique (Fig. 1) would alleviate the need to wait for the special stains to be available. This is especially useful for some urgent medical conditions such as crescentic glomerulonephritis or transplant rejection where quick and accurate diagnosis followed by rapid initiation of treatment may lead to significant improvements in clinical outcomes.

## Results

In order to prove the utility of our stain transformation technique, we investigated whether it can be used to improve preliminary diagnoses made by pathologists when only H&E is

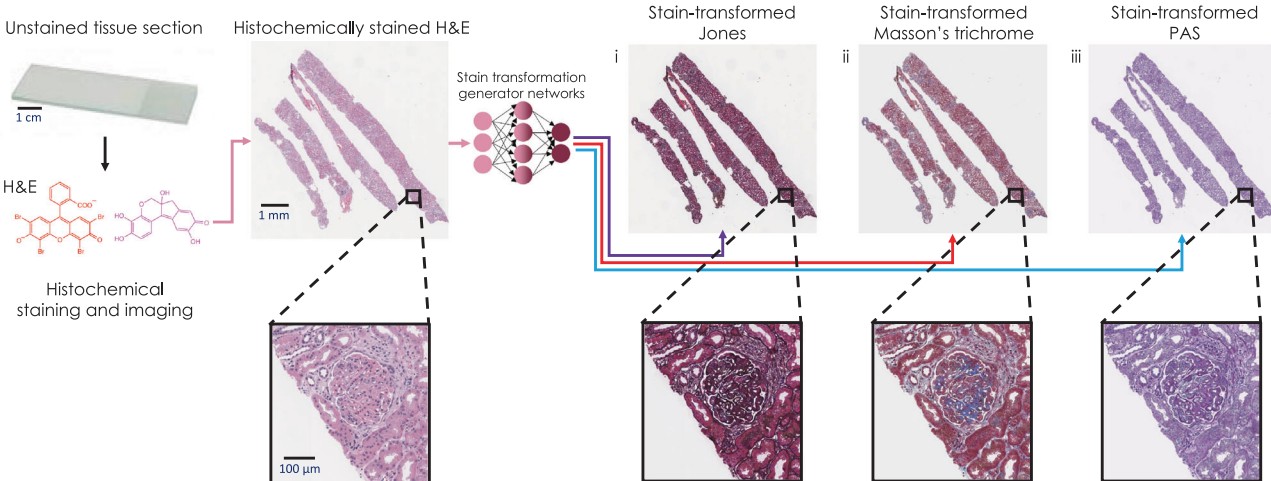

**Fig. 1 Overview of deep learning-based H&E stain transformation into special stains.** Histochemical staining of H&E is digitally transformed using a deep neural network into the special stains: (i) generation of JMS (purple arrow); (ii) generation of MT (red arrow); (iii) generation of PAS (blue arrow).

available. To do this, we used stain-to-stain transformation networks to create three additional computationally generated special stains, i.e., PAS, MT, and JMS, from existing H&E tissue sections. These WSIs were reviewed alongside the existing histochemically stained H&E images by pathologists (i.e., entirely bypassing the need to stain and wait for new slides). Based on tissue samples from 58 unique patients that are evaluated by three independent renal pathologists (i.e., $N = 174$ total diagnoses), our results revealed that the generation of the three stain-transformed special stains (PAS, MT, and JMS) improved the diagnoses in various nonneoplastic kidney diseases. These computationally generated panels of special stains transformed from existing H&E images using deep learning give the pathologists the additional information channels needed for a standard of patient care. We show that this unique stain-to-stain transformation workflow can be applied to a variety of diseases, and significantly improves the quality of the preliminary diagnosis when additional special stains are needed. We believe that this technique has significant utility in enhancing preliminary diagnoses, and could also provide time savings and help to reduce healthcare costs and burden for histopathology labs and patients.

**Design and training of stain transformation networks.** Deep neural networks were used to perform the transformation between H&E stained tissue and the special stains. To train these networks, a set of additional deep neural networks were used in conjunction with one another. This training workflow relies upon the ability for virtual staining of unlabeled tissue to generate images of different stains using a single unlabeled tissue section (Fig. 2a). By using a single neural network to generate both the H&E images alongside the special stains (PAS, MT, and JMS), a perfectly matched training image dataset can be created. However, due to the standardization of the output images generated using the staining network, the virtually stained images (to be used as inputs when training the stain transformation network) must be augmented with additional staining styles to ensure generalization. In other words, we designed our network to be able to handle inevitable variability in histochemical H&E staining that is a natural result of (i) differing staining procedures and reagents among histotechnologists and pathology labs and (ii) differences among digital WSI scanners that are being used. This augmentation is performed by $K = 8$ unique style transfer (staining normalization) networks (Fig. 2b), which ensured that a broad sample space is covered for the presented method to be

effective when applied to H&E stained tissue samples regardless of the inter-technician, inter-lab, or inter-equipment (e.g., WSI) variations observed at different institutions. Note here that these style transfer networks and the underlying training methods (e.g., CycleGANs) were solely used for H&E stain data augmentation. The use of CycleGANs only expands the sample space of the network inputs during the training, and their outputs were therefore not part of our stain transformation network loss function. This was possible since we utilized perfectly registered training images created by virtual staining of label-free autofluorescence images of tissue. This process simultaneously generated both the H&E and special stain images with a nanoscopic match in the local coordinates of each virtually stained image pair of our training dataset, which eliminated the need for the use of CycleGANs for stain-to-stain transformation.

Using this image dataset, the stain transformation network is trained, following the scheme shown in Fig. 2c. The network is randomly fed with image patches either coming from the virtually stained tissue, or the virtually stained images passing through one of the eight style transfer networks. The corresponding special stain (virtually stained from the same unlabeled field of view) is used as the ground truth regardless of the H&E style transfer. After its training, the network is then blindly tested on a variety of digitized H&E slides taken from UCLA repository, which represent a cohort of diseases and staining variations (all taken from patients that the network was not trained with). The network performs the stain transformation at the rate of ~1.5 mm²/s which takes in total ~0.5–1 min for a typical needle core kidney biopsy slide that was used in this study.

**Evaluation of stain transformation networks for kidney disease diagnoses.** To validate the presented stain transformation technique, a study was performed using WSI data from 58 different H&E stained tissue sections (each corresponding to a unique patient) obtained from an existing database of nonneoplastic kidney diseases. In this blinded study, three board-certified pathologists filled out diagnostic information for each H&E WSI (see the Methods section for details). Following a >3-week washout period, the same pathologists were asked to fill out the same diagnostic information, but along with the H&E, they were also provided the stain-transformed WSIs corresponding to special stains PAS, MT, and JMS, all generated from the existing H&E images. Following a second >3-week washout period, the pathologists were asked to fill out the same diagnostic

a) Virtual staining network (Generates stain transfer data)

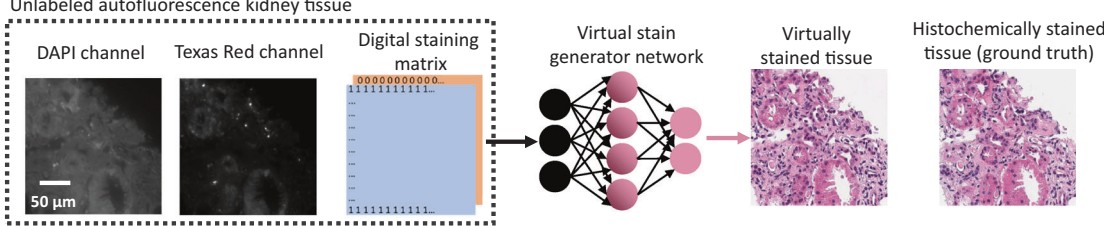

b) CycleGan style transfer network (Generates training inputs)

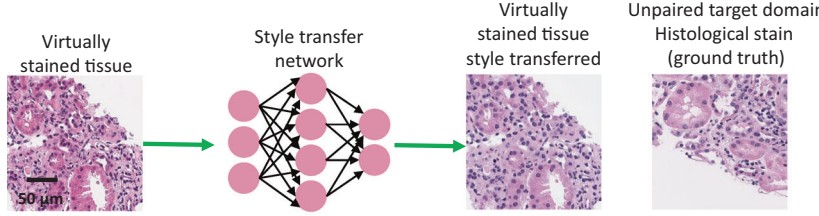

c) Stain transformation network training

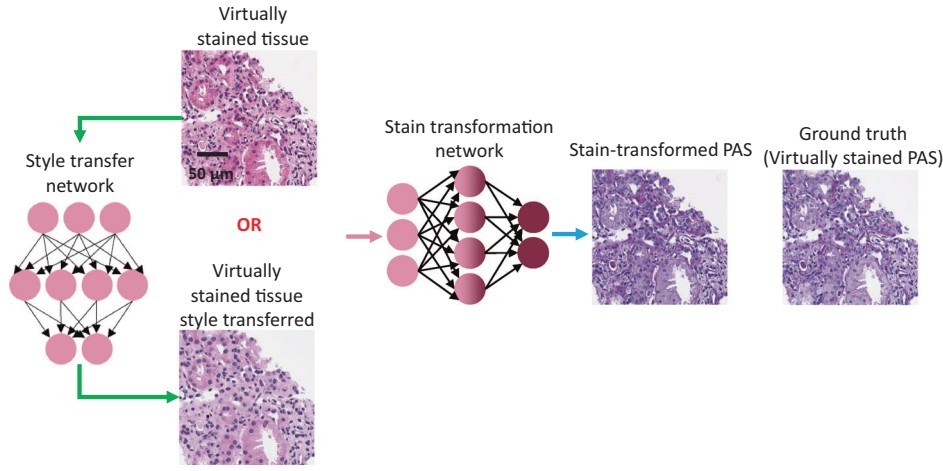

**Fig. 2 Deep neural networks used to generate the training data for the stain transformation network. a** Virtual staining network (pink arrow) which can generate both the H&E and special stain images. **b** Style transfer network (green arrow) that is used just to augment the training data. **c** Scheme used to train the stain transformation network. During its training, the stain transformation network is randomly given, as the input, either the virtually stained H&E tissue, or an image of the same field of view after passing through one of the eight style transfer networks. A perfectly matched virtually stained tissue image with the desired special stain (in this example shown: PAS) is used as the ground truth to train this neural network.

information. For this third phase, instead of using computationally generated, stain-transformed special stains, histochemically stained serial tissue sections were given to the pathologists along with the H&E (these sections originated from different depths within the tissue block). A diagram visualizing this study process can be seen in Fig. 3. Following the third round of diagnoses, a fourth board-certified pathologist adjudicated all the results/diagnoses and determined whether the viewing of the neural network generated special stains resulted in an Improvement (I), Concordance (C), or Discordance (D) with respect to the original H&E-only diagnoses. It is important to note that the official reported diagnosis that we used as our ground truth for this study also utilized additional information, such as electron microscopy and immunofluorescence images in order to make these diagnoses. The complete diagnostic information given by each pathologist, along with the adjudication for each report can be found in Supplementary Data 1.

Adjudication of the preliminary diagnoses made by using H&E only and the use of both H&E and stain-transformed special stains across the 58 cases revealed that using stain-to-stain transformations resulted in an average of 13 improved diagnoses (22.4%), 38.3 concordant diagnoses (66.1%), and 6.7 discordant diagnoses (11.5%) across the three pathologists. A total of ten cases had an improved preliminary diagnosis by two or more pathologists, while three cases had a discordant diagnosis by more than one pathologists (see Fig. 4). When comparing the diagnoses made with only H&E against those made with H&E alongside the histochemically stained special stains from serial tissue sections, an average of 15 improved diagnoses (25.8%), 38.6 concordant diagnoses (66.6%), and 4.3 discordant diagnoses (7.4%) were found across the three pathologists and 58 cases. For this second comparison, 12 cases were improved by two or more pathologists, while two cases were discordant for more than one pathologist (see Fig. 4).

## Phase 1
### Diagnosis based on H&E only

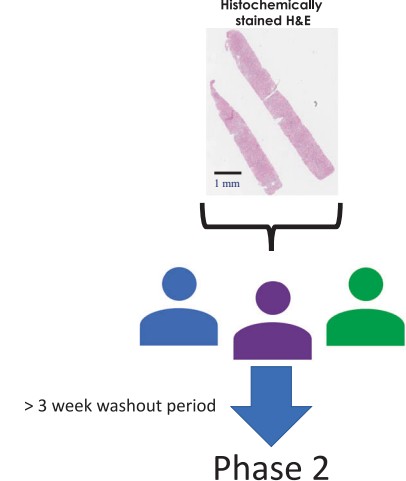

> 3 week washout period

## Phase 2
### Diagnosis based on H&E and stain-transformed special stains

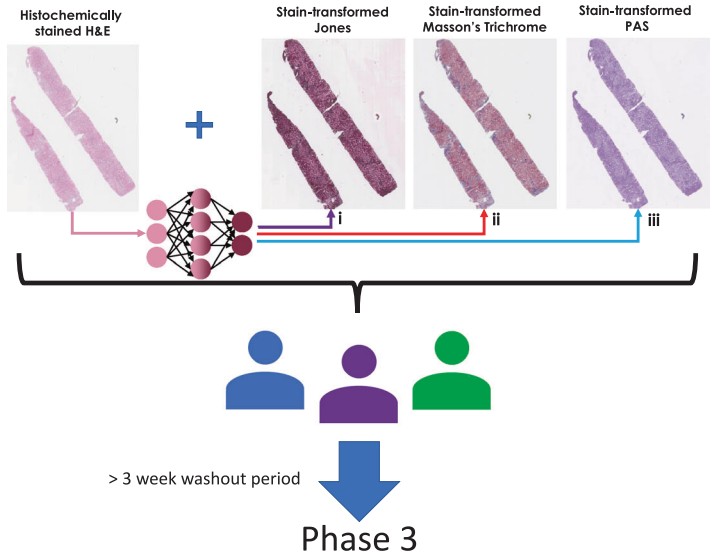

> 3 week washout period

## Phase 3
### Diagnosis based on H&E and histochemical special stains

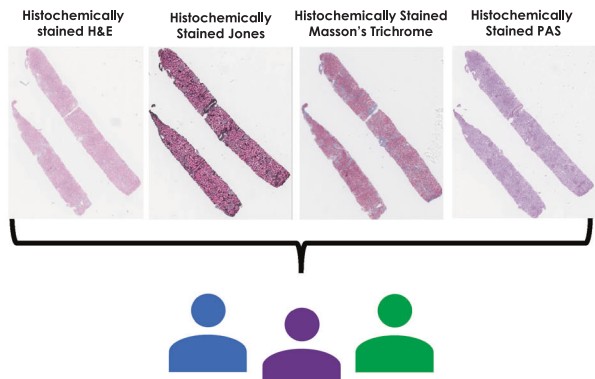

**Fig. 3 Overview of the study design.** Phase 1 shows the initial portion of the study where three pathologists review H&E WSIs of $N = 58$ different tissue sections (each from a unique patient). After a >3-week washout period, the second phase of diagnosis is performed, where the same three pathologists view the same WSIs, where, in addition to the H&E, the special stains generated by the stain transformation technique (PAS, Masson's Trichrome, Jones) are provided as well. After an additional >3-week washout period, the third phase of diagnosis is performed, where the same three pathologists again review the same WSIs. For this phase, instead of using special stains generated through the stain transformation technique, the images of all four stains (H&E, PAS, Masson's Trichrome, and Jones) come from histochemically stained serial sections. (i) Generation of JMS. (ii) Generation of MT. (iii) Generation of PAS.

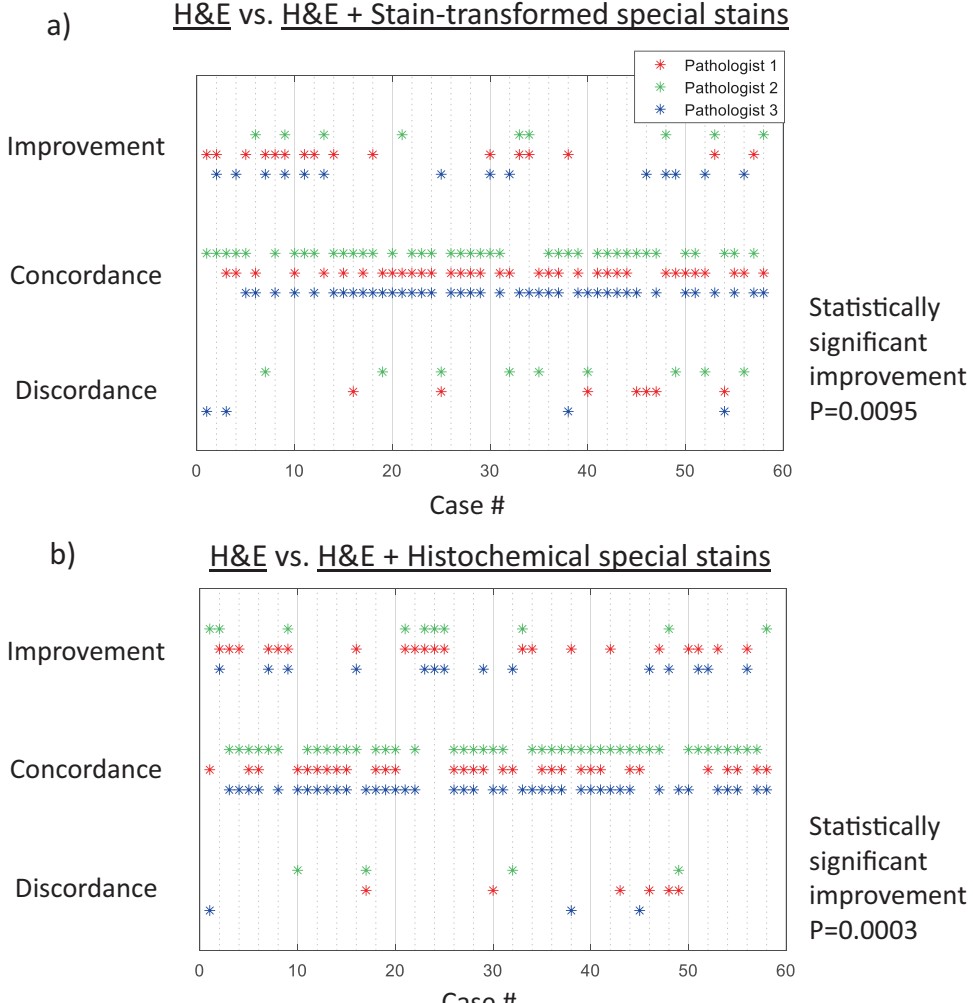

**Fig. 4 Visualization of the improvements, concordances, and discordances by case number for the two comparisons. a** Comparison of H&E only vs. H&E and the three stain-transformed special stains coming from the same tissue section. The use of the three stain-transformed special stains results in a statistically significant improvement over H&E only ($P = 0.0095$). **b** Comparison of H&E only vs. H&E and the three special stains (all histochemically stained) coming from serial tissue sections. The use of the three histochemically stained special stains results in a statistically significant improvement over H&E only ($P = 0.0003$). $P$ values were calculated using a one-sided $t$-test. No adjustments for multiple comparisons were needed.

These results show that the additional three virtual special stains improve the diagnostic outcome over a single histochemically stained H&E slide for a myriad of nonneoplastic diseases ($P = 0.0095$, using a one-tailed $t$-test). Our stain-to-stain transformation results are also in line with the level of improvement demonstrated when the pathologists had access to the H&E and three additional sections that are histochemically stained with the corresponding special stains ($P = 0.0003$, using a one-tailed $t$-test) over a single histochemically stained H&E slide. In addition to these, a secondary analysis was used to compare differences in the proportion of improvements, concordances, and discordances for each of these two comparisons reported in Fig. 4a, b. To do this, three separate chi-square tests were used— one for each pathologist (see the Methods section). These tests found that, while the histochemically stained tissue performed better for all three pathologists, the differences between the two comparisons were not statistically significant (with $P$ values of 0.60, 0.34, and 0.92 for the first, second, and third pathologist, respectively).

For each of the diagnoses marked as improvements, the pathologists were able to provide more accurate characterization or a more complete diagnosis. As an example, Fig. 5 demonstrates the improvement using the presented stain transformation technique for a case used for the preliminary evaluation of our technique (diagnosed with acute cellular rejection and acute antibody-mediated rejection), where all three pathologists had the quality of their diagnoses improved. These improvements appear to be based on the clearer definition of the tubular and glomerular basement membranes in the computationally generated special stains. This biopsy contains very pronounced cellular inflammation that is difficult to precisely localize on a standard H&E stain, as H&E does not give clear contrast to structures such as basement membranes. The computationally generated special stains highlight the tubular basement membranes which allowed all three pathologists to see the location of the inflammatory cells and give a more precise characterization of the organ rejection process. Another example is case #2, where two pathologists were able to provide a diagnosis of membranous nephropathy only after a review of the stain-transformed JMS stain, which is demonstrated in Fig. 6a. In this case, the generated JMS helped the visualization of changes to the basement membrane which are characteristic of membranous nephropathy.

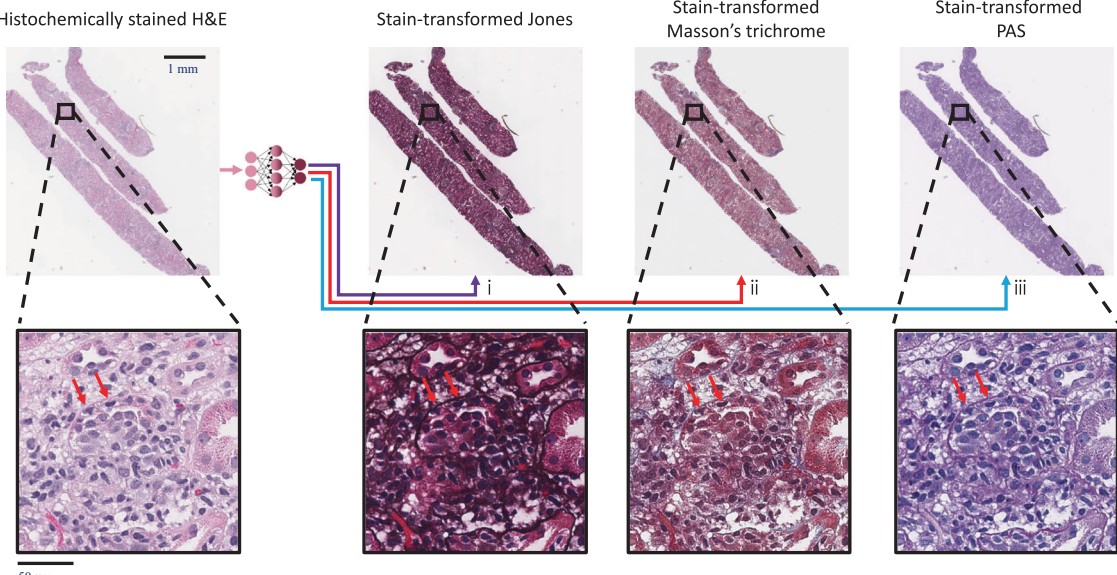

**Fig. 5 Examples of improved diagnoses fostered by the stain-transformed special stains.** We report here WSIs that are generated using the stain transformation technique. In this case, the addition of the computationally generated special stains improved all three of the diagnoses made by the pathologists. The red arrows point to a region, where the special stains help highlight inflammatory cells within the tubule, otherwise, the boundary of the tubules cannot be seen with the H&E stain only. (i) Generation of JMS. (ii) Generation of MT. (iii) Generation of PAS. A total of 58 cases were viewed by three pathologists to perform the statistical analysis.

The discordances were broken up into two categories: those which were determined to be due to pathologist interpretation error (e.g., case #7), and those which are likely due to misrepresentation of the image on the virtual stains (see Supplementary Data 1 for details). As an example, in case #1, the fibrin thrombi in a case of thrombotic microangiopathy (TMA) appeared too pale on the stain-transformed PAS stain. An example field of view (FOV) with the matching histochemically stained FOV from an adjacent serial tissue section can be seen in Fig. 6b. As a second example, in case #3 (amyloidosis), amyloid deposits were darker on the stain-transformed JMS stain than would be typical in histologically stained slides (an example FOV can be seen in Supplementary Fig. 1). It is worth emphasizing that in both of these cases (#1 and #3), two of the three pathologists were able to make concordant diagnoses. Furthermore, one pathologist made a more definitive diagnosis of TMA with the aid of the stain-transformed special stains in case #1 in addition to the original images of the histochemically stained H&E.

We should note that previous research on statistical evaluation of intra-observer decisions revealed a small intra-observer disagreement rate of ~4% when the same cases are viewed by the same pathologist at two different time points[22]. This could potentially account for the discordance in some of the cases such as #7, which was determined to be due to pathologist interpretation error.

**Evaluation of the quality of stain-transformed special stain images.** An additional study was performed to assess the quality of the stains generated by the stain transformation network. For this study, three pathologists rated the quality of various aspects of the stains generated using the stain transformation network as well as the images of histochemically stained tissue from serial tissue sections. The pathologists each viewed 16 unique rectangular FOVs (with dimensions ranging from ~150 μm × 175 μm to ~375 μm × 500 μm) coming from the three validation slides used during the training of our neural network. These same FOVs were scored for each of the three generated special stains, as well as for the same region of the tissue in a serial histochemically stained

tissue section. The 16 FOVs were randomly chosen by one of the pathologists to be representative of the tissue sections used, and only in-focus areas where the tissue is unbroken in all of the tissue sections were selected. The FOVs were ordered randomly, and each pathologist rated every image twice—before and after the image randomly being rotated or flipped (resulting in a total of 32 ratings for each stain and image type). Overall, each one of the three pathologists rated 192 FOVs (half virtually stained and half histologically stained).

The pathologists scored four aspects of each FOV on a scale from 1 to 4, where 4 is perfect, 3 is very good, 2 is good enough (passable), and 1 is not acceptable, for a total of 2304 unique assessments/ratings made by three pathologists. The MT stain was rated for overall stain quality, nuclear detail, cytoplasmic detail, and extracellular fibrosis quality. The PAS and Jones Silver stains were rated based on their overall stain quality, along with nuclear detail, cytoplasmic detail, and basement membrane detail.

Supplementary Table 1 shows the mean score for each stain type and quality metric. This table shows that the difference in stain quality for all of the measured aspects of each stain is significantly smaller than the standard error between the ratings. This indicates that the stain-to-stain transformation technique achieves the quality of stain equivalent to that of the histochemically stained tissue used as our ground truth. A non-randomized version of the images used for this stain quality evaluation study can be found in the Supplementary Data 2 file.

## Discussion

While different approaches have been explored over the past few years to perform a transformation between two stains, the approach presented here has several unique advantages: (1) it involves less chemical processing applied to tissue, without the need for destaining and re-staining; and (2) our approach is based on supervised training of the stain transformation network using pairs of perfectly registered training images that are created by label-free virtual staining, which constitutes a precise structural fidelity constraint for the distribution loss that is learned by the discriminator, significantly helping its generalization. Stated

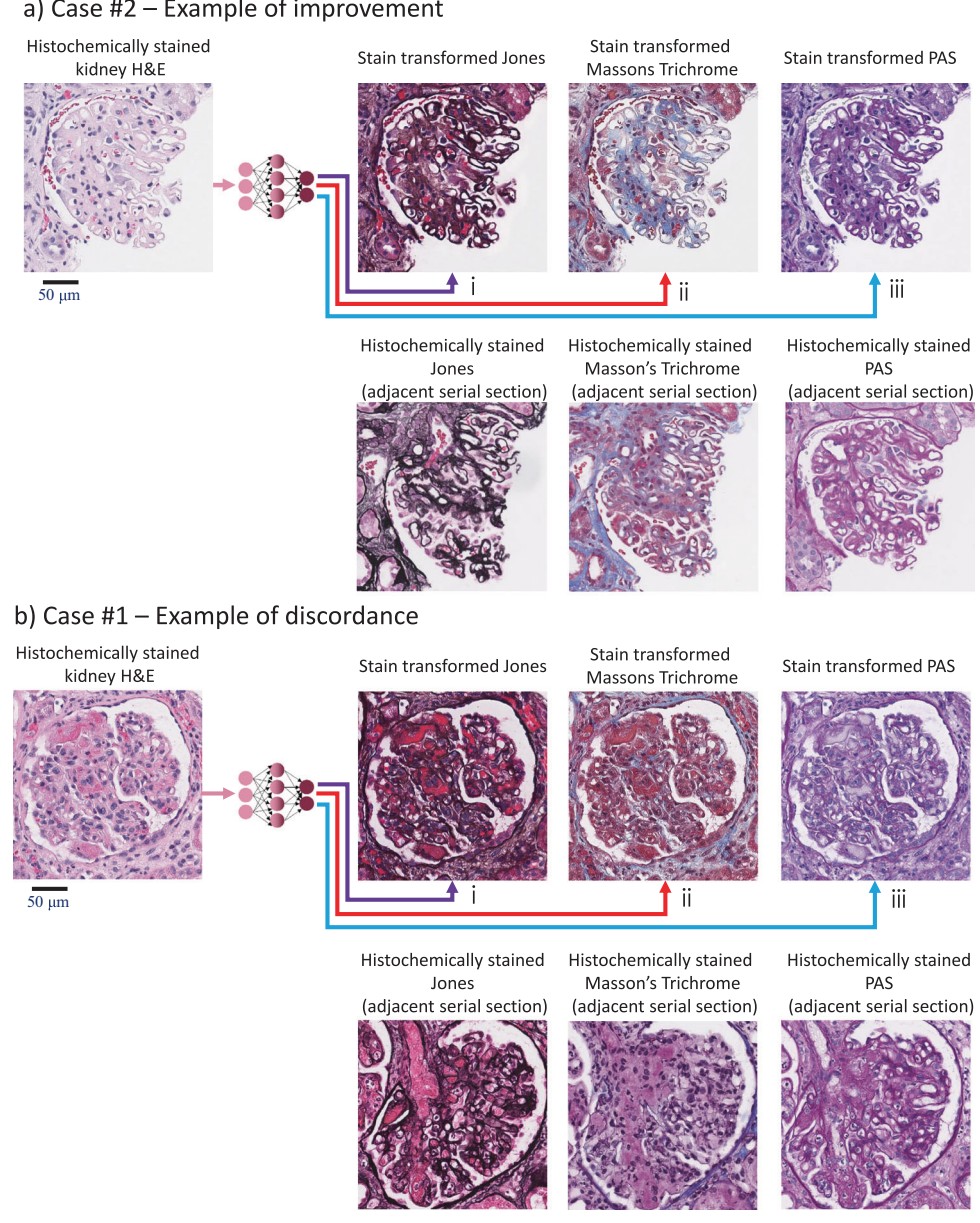

**Fig. 6 Examples of improved and discordant diagnosis achieved by the stain-transformed special stains. a** Example of improved diagnosis fostered by the stain-transformed special stains. For case #2 (in Fig. 4 and the Supplementary Data 1), the basement membrane changes that are characteristic of membranous nephropathy (subepithelial spikes and basement membrane holes) are only appreciated after reviewing the stain-transformed JMS. The bottom images exemplify histochemically stained images of adjacent serial sections of the patient sample; that is why they correspond to different sections within the tissue block. **b** Example of the discordance demonstrated between the H&E and computationally generated special stains for case #1 (in Fig. 4 and the Supplementary Data 1). In this field of view, the fibrin thrombi are gray-yellow in color on the stain-transformed PAS stain rather than pink-red. (i) Generation of JMS. (ii) Generation of MT. (iii) Generation of PAS. A total of 58 cases were viewed by three pathologists to perform the statistical analysis.

differently, no stain-to-stain image aberrations or misalignments exist in this training data due to the fact that the source of information (autofluorescence of the label-free tissue) is common for all the virtually stained images. This feature significantly improves the reliability and accuracy of the stain-to-stain transformation that is learned using our method. These important advantages are enabled by using autofluorescence-based virtual staining of label-free tissue sections with multiple stains to create perfectly paired training image datasets. While in this paper we used autofluorescence to generate contrast from label-free tissue, other contrast mechanisms such as quantitative phase imaging, multi-photon-microscopy, fluorescence lifetime imaging, and photoacoustic microscopy, among others, can also support this

supervised training of the presented stain transformation method. The resulting networks that are trained with our methodology can digitally transform any existing chemically stained tissue image into new types of stains.

Similar to the validation of digital pathology systems in general, a perfect stain-to-stain transformation is not required; the standard of practice is to demonstrate a lack of inferiority, which is what we have endeavored to do. Substituting chemically stained slides for stain-transformed slides leads to several advantages, including e.g., decreased slide preparation time, decreased laboratory costs, and preservation of tissue for subsequent analysis, if necessary. In the future, the use of computationally generated special stains may make it possible to selectively omit the need for performing actual

special stains and save time and laboratory expenses in some settings.

The ability of this stain-to-stain transformation network to generalize across stain variations is also highly beneficial as there are significant differences among stains produced by different labs and even across stains performed by the same histotechnician (e.g., Supplementary Fig. 2a demonstrates three examples of such variations for stains produced by the same lab). However, in order for a stain transformation technique to be effective for any practical application, the network must generalize across this wide sample space. As one of the key features of virtual staining is stain normalization[7], the network requires data augmentation to better facilitate the learning across a wide input staining distribution. For this purpose, we used a set of eight CycleGAN networks to perform this stain data augmentation of the H&E dataset used to train our stain transformation network. The use of CycleGAN networks to perform a stain normalizing style transfer has been shown to be more effective than traditional stain normalization algorithms[23]. Furthermore, they have proven to be highly effective at performing data augmentation for medical imaging[24]. By applying these CycleGAN augmentation networks to our training image dataset, we were able to successfully generalize to various slides used for blind testing. Three examples of this CycleGAN-based stain augmentation results are reported in Supplementary Fig. 2b, which demonstrates that the three different networks are capable of converting the virtually stained tissue to have H&E distributions which match the distributions seen in Fig. S2a. Furthermore, the results show that the same stain transformation network is consistent across these various distributions as there is little variation among the virtual PAS outputs (Supplementary Fig. 2b). These style normalization/transfer networks used in data augmentation can be easily further expanded upon, if needed, using existing databases of H&E images.

As we have emphasized earlier, these style transfer networks were only used for H&E stain data augmentation and were not included in our stain transformation loss function. We utilized perfectly registered training images generated by virtual staining of label-free tissue; as a result of this, potential hallucinations or artifacts related to unsupervised training with CycleGANs and unpaired training data are eliminated (as can be seen in Supplementary Fig. 3). When the same CycleGAN architecture used for the data augmentation is applied to the various stain transformations, a number of clear hallucinations occur. These hallucinations are particularly evident for the PAS and Jones Silver stain, where the networks incorrectly label the tubular basement membranes (see Supplementary Fig. 3). The tubules are composed of epithelial cells lining basement membranes that stain black on the Jones stain and magenta on the PAS stain. The brush border lining the luminal surface of the epithelial cells is also normally lightly stained black and magenta by the Jones and PAS stains, respectively. The CycleGAN method incorrectly recognized the basement membranes and tubular brush borders leading to incorrect image generation, which is a very significant error. In contrast, the quality and features of the MT stain appear to be more similar between the two techniques. This is believed to be due to the MT stain being relatively similar to H&E, while the other stains require significant structural changes which can cause hallucinations for CycleGANs. These results and observations highlight the significant advantages of our stain-to-stain transformation network compared to standard CycleGAN-based methods.

It is important to note that the current stain-to-stain network is trained to work with H&E stains performed at a few institutions and imaged by different microscopes from the same vendor/model (Leica Biosystems Aperio AT2 slide scanner). Additional data would be required for the network to generalize to samples imaged using microscopes with different specifications or vendors, or any H&E stains which are performed in a significantly different manner. Furthermore, while this study covers a broad range of diseases, it is still a proof of concept. Future studies should be performed which contain both larger training and test datasets in order to conclusively show the technique may be suitable for diagnostic use. Future work may also apply this technique that we have presented to other biomarkers that are currently labeled with IHC to help target specific conditions.

In addition to histological stains, immunofluorescence and electron microscopy[25] based evaluation play significant roles in the standard of care for nonneoplastic kidney biopsy evaluation. In this study, we have attempted to isolate the role of standard light microscopy in the nonneoplastic kidney disease evaluation and therefore these other modalities were not included. However, their application in clinical cases would only serve to support the pathologic final diagnosis and add a layer of further confirmation and safety to this resource-saving stain transformation technique.

In this work, we focused on image transformations from H&E to special stains, since H&E is used as the bulk of the staining procedures, covering ~80% of all the human tissue staining procedures[1]. However, other stain-to-stain transformations can also be considered. For example, transformations from special stains to H&E or from immunofluorescence to H&E or special stains could be performed using the presented method. Our approach allows pathologists to visualize different tissue constituents without waiting for additional slides to be stained with special stains, and we demonstrated it to be effective for the clinical diagnosis of multiple renal diseases. Another advantage of the presented technique is that it can rapidly perform the stain transformation (at a rate of 1.5 mm²/s on a consumer-grade desktop computer with two GPUs), while saving labor, time, chemicals, and can significantly benefit the patient as well as the healthcare system.

## Methods

**Training of stain transformation network.** All of the stain transformation networks and virtual staining networks used in this paper were trained using GANs. Each of these GANs consists of a generator ($G$) and a discriminator ($D$). The generator is used to perform the transformation of the input images ($x_{input}$), while the discriminator network is used to help train the network to generate images, which match the distribution of the ground truth stained images. It does this by trying to discriminate between the generated images ($G(x_{input})$) and the ground truth images ($z_{label}$). The generator is in turn taught to generate images, which cannot be classified correctly by the discriminator. This GAN loss is used in conjunction with two additional losses: a mean absolute error ($L_1$) loss and a total variation (TV) loss. The $L_1$ loss is used to ensure that the transformations are performed accurately in space and color, while the TV loss is used as a regularizer, and reduces noise created by the GAN loss. Together, the overall loss function is described as:

$$l_{generator} = L_1\{z_{label}, G(x_{input})\} + \alpha \times TV\{G(x_{input})\} + \beta \times (1 - D(G(x_{input})))^2 \quad (1)$$

where $\alpha$ and $\beta$ are constants used to balance the various terms of the loss function. The stain transformation networks are tuned such that the $L_1$ loss makes up ~1% of the overall loss, the TV loss makes up only ~0.03% of the overall loss, and the discriminator loss makes up the remaining ~99% of the loss (relative ratios change over the course of the training). The $L_1$ portion of the loss can be written as:

$$L_1(z,G) = \frac{1}{P \times Q} \sum_p \sum_q |z_{p,q} - G(x_{input})_{p,q}| \quad (2)$$

where $p$ and $q$ are the pixel indices and $P$ and $Q$ are the total number of pixels in each image. The total variation loss is defined as:

$$TV(G(x_{input})) = \sum_p \sum_q |G(x_{input})_{p+1,q} - G(x_{input})_{p,q}| + |G(x_{input})_{p,q+1} - G(x_{input})_{p,q}| \quad (3)$$

The discriminator network has a separate loss function which is defined as:

$$l_{discriminator} = D(G(x_{input}))^2 + (1 - D(z_{label}))^2 \quad (4)$$

A modified U-net[1] neural network architecture was used for the generator, while the discriminator used a VGG-style[2] network. The U-net architecture uses a

set of four up-blocks and four down-blocks, each containing three convolutional layers with a $3 \times 3$ kernel size, activated upon by the LeakyReLU activation function which is described as:

$$\text{LeakyReLU}(x) = \begin{cases} x & \text{for } x > 0 \\ 0.1\, x & \text{otherwise} \end{cases} \quad (5)$$

The first down-block increases the number of channels to 32, while the rest each increase the number of channels by a factor of two. Each of these down-blocks ends with an average pooling layer which has both a stride and a kernel size of two. The up-blocks begin with a bicubic up-sampling prior to the application of the convolutional layers. Between each of the blocks of a certain layer, a skip connection is used to pass data through the network without needing to go through all the blocks. After the final up-block, a convolutional layer maps back to three channels.

The discriminator is made up of five blocks. These blocks contain two convolutional layers and LeakyReLU pairs, which together increase the number of channels by a factor of two. These are followed by an average pooling layer with a stride of two. After the five blocks, two fully connected layers reduce the output dimensionality to a single value, which in turn is input into a sigmoid activation function to calculate the probability that the input to the discriminator network is real, i.e., not generated.

Both the generator and discriminator were trained using the adaptive moment estimation (Adam)[26] optimizer to update the learnable parameters. A learning rate of $1 \times 10^{-5}$ was used for the discriminator network while a rate of $1 \times 10^{-4}$ was used for the generator network. For each iteration of the discriminator training, the generator network is trained for seven iterations. This ratio reduces by one every 4000 iterations of the discriminator to a minimum of one discriminator iteration for every three generator iterations. The network was trained for 50000 iterations of the discriminator, with the model being saved every 1000 iterations. The best generator model was chosen manually from these saved models by visually comparing different models. For all three of the generator networks (MT, PAS, and JMS), the 15,000th iteration of the discriminator was chosen as the optimal model.

The stain transformation networks were trained using pairs of $256 \times 256$-pixel image patches generated by the class conditional virtual staining network (label-free), downsampled by a factor of 2 (to match 20× magnification). These patches were randomly cropped from one of 1013 $712 \times 712$-pixel images coming from ten unique tissue sections, leading to ~7836 unique patches usable for training. Seventy-six additional images coming from three unique tissue sections were used to validate the network. These images were augmented using the eight stain augmentation networks and further augmented through random rotation and flipping of the images. The diagnoses of each of the samples used for training and validation have been added to Supplementary Tables 2 and 3. Each of the three stain transformation networks (MT, PAS, and JMS) were trained using images generated by the label-free virtual staining networks from the same input autofluorescence images. Furthermore, the images were converted to the YCbCr color space[27] before being used as either the input or ground truth for the neural networks.

As this stain transformation neural network performs an image-to-image transformation, it learns to transform specific structures using the ~513 million pixels in the dataset that are independently accounted for in the loss function. Furthermore, since the network learns to convert structures which are common throughout many different types of samples, it can be applied to tissues with diseases that the network was not trained with. When used in conjunction with the eight data augmentation networks which convert the values of these pixels, as well as random rotation and flipping (for an additional 8×) augmentation, there are effectively many billions of pixels which are used to learn the desired stain-to-stain transformation. Because of these advantages, a much smaller number of training samples from unique patients can be used than would be required for a typical classification neural network.

**Image data acquisition**. All of the neural networks were trained using data obtained by microscopic imaging of thin tissue sections coming from needle core kidney biopsies. Unlabeled tissue sections were obtained from the UCLA Translational Pathology Core Laboratory (TPCL) under UCLA IRB 18-001029, from an existing specimen. The autofluorescence images were captured using an Olympus IX-83 microscope (controlled with the MetaMorph microscope automation software, version 7.10.161), using a DAPI filter cube (Semrock OSFI3-DAPI5060C, EX 377/50 nm EM 447/60 nm) as well as a Texas Red filter cube (Semrock OSFI3-TXRED-4040C, EX 562/40 nm EM 624/40 nm) to generate the second autofluorescence image channel.

In order to create the training dataset for the virtual staining network, pairs of matched unlabeled autofluorescence images and brightfield images of the histochemically stained tissue were obtained. H&E, MT, and PAS histochemical staining were performed by the Tissue Technology Shared Resource at UC San Diego Moores Cancer Center. The JMS staining was performed by the Department of Pathology and Laboratory Medicine, Cedars-Sinai Medical Center, Los Angeles, CA, USA. These stained slides were digitally scanned using a brightfield scanning microscope (Leica Biosystems Aperio AT2 slide, using 40x/0.75NA objective). All the slides and digitized slide images were prepared from an existing specimen. Therefore, this work did not interfere with standard practices of care or sample collection procedures. The H&E image dataset used for the study came from the

existing UCLA pathology database containing WSIs of stained kidney needle core biopsies, under UCLA IRB 18-001029. These slides were similarly imaged using Aperio AT2 slide scanning microscopes.

**Image co-registration**. To train label-free virtual staining networks, the autofluorescence images of unlabeled tissue were co-registered to brightfield images of the same tissue after it had been histochemically stained. This image co-registration was done through a multistep process[28], beginning with a coarse matching which was progressively improved until subpixel level accuracy is achieved. The registration process first used a cross-correlation-based method to extract the most similar portions of the two images. Next, the matching was improved using multimodal image registration[29]. This registration step applied an affine transformation to the images of the histochemically stained tissue to correct for any changes in size or rotations. To achieve pixel-level co-registration accuracy, an elastic registration algorithm was then applied. However, this relies upon a local correlation-based matching. Therefore, to ensure that this matching could be accurately performed, an initial rough virtual staining network is applied to the autofluorescence images[7,8]. These roughly stained images were then co-registered to the brightfield images of the stained tissue using a correlation-based elastic pyramidal co-registration algorithm[30].

Once the image co-registration is complete, the autofluorescence images were normalized by subtracting the average pixel value of the tissue area for the WSI and subsequently dividing it by the standard deviation of the pixel values in the tissue area.

**Class conditional virtual staining of label-free tissue**. A class conditional GAN was used to generate both the input and the ground truth images to be used during the training of the presented stain transformation networks (Fig. 2a). This class conditional GAN allows multiple stains to be created simultaneously using a single deep neural network[8]. To ensure that the features of the virtually stained images are highly consistent between stains, a single network must be used to generate the stain transformation network input (virtual H&E) and the corresponding ground truth images (virtual special stains) that are automatically registered to each other as the information source is the same image. This is only required for the training of the stain transformation neural networks and is rather beneficial as it allows both the H&E and special stains to be perfectly matched. Furthermore, an alternative image dataset made up of co-registered virtually stained and histochemically stained fields of view will present limitations due to imperfect co-registration and deformities caused by the staining process. These are eliminated by using a single class conditional GAN to generate both the input and the ground truth images.

This network uses the same general architecture as the network described in the previous section, with the addition of a Digital Staining Matrix concatenated to the network input for both the generator and discriminator[8]. This staining matrix defines the stain coordinates within a given image FOV. Therefore, the loss functions for the generator and discriminator are:

$$l_{\text{generator}} = L_1\{z_{\text{label}}, G(x_{\text{input}}, \widetilde{c})\} + \alpha \times \text{TV}\{G(x_{\text{input}}, \widetilde{c})\} + \beta \times (1 - D(G(x_{\text{input}}, \widetilde{c}), \widetilde{c}))^2 \quad (6)$$

$$l_{\text{discriminator}} = D(G(x_{\text{input}}, \widetilde{c}), \widetilde{c})^2 + (1 - D(z_{\text{label}}, \widetilde{c}))^2 \quad (7)$$

where $\widetilde{c}$ is a one-hot encoded digital staining matrix with the same pixel dimensions as the input image. When used in the testing phase, the one-hot encoding allows the network to generate two separate stains (H&E and the corresponding special stain) for each FOV.

The number of channels in each layer used by this deep neural network was increased by a factor of two compared to the stain transformation architecture described above to account for the larger dataset size and the need for the network to perform two distinct stain transformations.

A set of four adjacent tissue sections were used to train the virtual staining networks for H&E and the three special stains. The H&E portion of all three of the networks was trained with 1058 $1424 \times 1424$-pixel images coming from ten unique patients, the PAS network was trained with 946 $1424 \times 1424$-pixel images coming from 11 unique patients, the Jones network was trained with 816 $1424 \times 1424$-pixel images coming from ten unique patients, and the MT network was trained with 966 $1424 \times 1424$-pixel images coming from ten unique patients. A list of the samples used to train the various networks, and the original diagnoses of the patients can be seen in Supplementary Table 2. All of the stains were validated using the same three validations slides.

**Style transfer for H&E image data augmentation**. In order to ensure that the stain transformation neural network is capable of being applied to a wide variety of histochemically stained H&E images, we use the CycleGAN[18] model to augment the training dataset by performing style transfer (Fig. 2b). As discussed, these CycleGAN networks only augment the image data used as inputs in the training phase. This CycleGAN model learns to map between two domains $X$ and $Y$ given the training samples $x$ and $y$, where $X$ is the domain for the original virtually stained H&E and $Y$ is the domain for the H&E image generated by a different lab or hospital. This model performs two mappings $G : X \rightarrow Y$ and $F : Y \rightarrow X$. In addition, two adversarial discriminators $D_X$ and $D_Y$ are introduced. A diagram showing the relationship between these various networks is shown in Supplementary Fig. 4.

The loss function of the generator $l_{generator}$ contains two types of terms: adversarial losses $l_{adv}$ to match the stain style of the generated images to the style of histochemically stained images in target domain; and cycle consistency losses $l_{cycle}$ to prevent the learned mappings $G$ and $F$ from contradicting each other. The overall loss is therefore described by:

$$l_{generator} = \lambda \times l_{cycle} + \varphi \times l_{adv} \tag{8}$$

where $\lambda$ and $\varphi$ are relative weights/constants. For each of the networks, we set $\lambda = 10$ and $\varphi = 1$. Each generator is associated with a discriminator, which ensures that the generated image matches the distribution of the ground truth. The adversarial losses for each of the generator networks can be written as:

$$l_{advX \to Y} = \left(1 - D_Y(G(x))\right)^2 \tag{9}$$

$$l_{advY \to X} = \left(1 - D_X(F(y))\right)^2 \tag{10}$$

And the cycle consistency loss can be described as:

$$l_{cycle} = L_1\{y, G(F(y))\} + L_1\{x, F(G(x))\} \tag{11}$$

The adversarial loss terms used to train $D_X$ and $D_Y$ are defined as:

$$l_{D_X} = \left(1 - D_X(x)\right)^2 + D_X(F(y))^2 \tag{12}$$

$$l_{D_Y} = \left(1 - D_Y(y)\right)^2 + D_Y(G(x))^2 \tag{13}$$

For these CycleGAN models, $G$ and $F$ use U-net architectures similar to the stain transformation network. It consists of three down-blocks followed by three up-blocks. Each of these down-blocks and up-blocks are identical to the corresponding blocks in the stain transformation network. $D_X$ and $D_Y$ also have similar architectures to the discriminator network of stain transformation network. However, they have four blocks rather than five blocks as in the previous model.

During the training, the Adam optimizer was used to update the learnable parameters with learning rates of $2 \times 10^{-5}$ for both the generator and discriminator networks. For each step of discriminator training, one iteration of training was performed for the generator network, and the batch size for training was set to 6.

A list of the original diagnoses of the samples used to train the CycleGAN stain augmentation networks can be seen in Supplementary Table 3. The same table also indicates how many FOVs were used for each sample used to train the CycleGAN network.

**Training of single-stain virtual staining networks**. In addition to performing multiple virtual stains using a single neural network, separate networks which only generate one individual virtual stain each were also trained. These networks were used to perform the rough virtual staining that enables the elastic co-registration. These networks use the same general architecture as the stain transformation networks, with the only difference being that the first block in both the generator and the discriminator increases the number of channels to 64. The input and output images are the autofluorescence images and the histochemically stained images, respectively, processed using the image registration described in the image co-registration section.

**Implementation details**. The image co-registration was implemented in MATLAB using version R2018a (The MathWorks Inc.). The neural networks were trained and implemented using Python version 3.6.2 with TensorFlow version 1.8.0. The timing was measured on a Windows 10 computer with two Nvidia GeForce GTX 1080 Ti GPUs, 64GB of RAM, and an Intel I9-7900X CPU.

**Pathologic evaluation of kidney biopsies**. An initial study of 16 sections—comparing the diagnoses made with H&E only against the diagnoses made with H&E as well as the stain-transformed special stains—was first performed to determine the feasibility of the technique. For this initial evaluation, 16 non-neoplastic kidney cases were selected by a board-certified kidney pathologist (J.E.Z.) to represent a variety of kidney diseases (listed in Supplementary Data 1). For each case, the WSI of the histochemically stained H&E slide, along with a worksheet that included a brief clinical history, were presented to three board-certified renal pathologists (W.D.W, M.F.P.D., and A.E.S.). The diagnostic worksheet can be seen in Supplementary Table 4. The WSIs were exported to the Zoomify format[31], and uploaded to the GIGAmacro[32] website to allow the pathologists to confidentially view the images using a standard web browser. The WSIs were viewed using standard displays (e.g., LCD Monitor, FullHD, 1920 × 1080 pixels).

In the diagnostic worksheet, the reviewers were given the H&E WSI and brief patient history and asked to make a preliminary diagnosis and quantify certain features of the biopsy (i.e., number of glomeruli and arteries) and provide additional comments if necessary. After a >3-week washout period to reduce the pathologists' familiarity with the cases, the three reviewing pathologists received, in addition to the same histologically stained H&E WSIs and the same patient medical history, three computationally generated special stain WSIs for each case: MT, PAS, and JMS. Being given these slides, they were asked to provide a preliminary diagnosis for a second time. This >3-week washout period was chosen to be 1 week

greater than the College of American Pathologists Pathology and Laboratory Quality Center guidelines[33], ensuring that the pathologists were not influenced by previous diagnoses.

To test the hypothesis that using additional stain-transformed WSIs can be used to improve the preliminary diagnosis, the adjudicator pathologist (J.E.Z.) who was not among the three diagnosticians provided judgment to determine Concordance (C), Discordance (D), or Improvements (I) between the diagnosis quality of the first and second round of preliminary diagnoses provided by the group of diagnosticians (see Supplementary Table 4).

To expand the total number of cases to 58 and perform the third study, (Fig. 3) the same set of steps were repeated. To allow for higher throughput, in this case, the WSIs were uploaded to a custom-built online file viewing server based on the Orthanc server package[34]. Using this online server, the user is able to swap between the various cases. For each case, the patient history is presented, along with the WSI and the option to swap between the various stains, where applicable. The pathologists were asked to input their diagnosis, the chronicity, and any comments that they might have into text boxes within the interface.

Once the pathologists completed the diagnoses with H&E only as well as with H&E and the stain-transformed special stains, another >3-week washout period was observed. Following this second washout period, the pathologists were given WSIs of the original histochemically stained H&E along with the three histochemically stained special stains coming from serial tissue sections. Two of these cases used in the preliminary study were excluded from the final analysis, as WSIs of the three special stains could not be obtained from serial tissue sections. For the first of these excluded cases, all of the pathologist's diagnoses were improved using stain-to-stain transformation, and for the second, one of the diagnoses was improved while the other two pathologists' diagnoses were concordant.

The pathologists' diagnoses and comments can be found in the file Supplementary Data 1. Pathologist 2 was replaced for the expanded study due to time availability. Therefore, there is a separate page containing the initial study diagnoses for this pathologist.

**Statistical analysis**. Using the preliminary study of 16 samples, we calculated that a total of 41 samples are needed to show statistical significance (using a power of 0.8 and an alpha level of 0.05 and using a one-tailed $t$-test). Therefore, the total number of patients was increased to 58 to ensure that the study was sufficiently powered.

A one-tailed $t$-test was used to determine whether a statistically significant number of improvements were made when using either [H&E and stain-transformed special stains], or [H&E and histochemically stained special stains] over only [H&E] images. The statistical analyses were performed by giving a score of $+1$ to any improvement, $-1$ to any discordance, and 0 to any concordance. The score for each case was then averaged among the three pathologists who evaluated the case, and the test showed that the amount of improvement (i.e., if the average score is greater than zero) across the 58 cases was statistically significant.

A chi-squared test with two degrees of freedom was used to compare the proportion of improvements, concordances, and discordances between the methods tested above. The improvements, concordances, and discordances for each pathologist was compared individually.

For all tests, a $P$ value of 0.05 or less was considered to be significant.

**Reporting Summary**. Further information on research design is available in the Nature Research Reporting Summary linked to this article.

## Data availability

Data supporting the results demonstrated by this study are available within the main text and the Supplementary Information. The full set of images used for the stain quality assessment study can be found in the Supplementary Data 2 file as well as at: https://github.com/kevindehaan/stain-transformation. The full pathologist reports and adjudication results can be found in the Supplementary Data 1 file and on GitHub. Examples of patient sample fields of view can also be found on GitHub at https://github.com/kevindehaan/stain-transformation. For each example, the histochemically stained H&E and stain transformed special stains are shown for the same FOV, while the histochemically stained special stains that are shown come from the same biopsy, through serial tissue sections. Raw whole slide images corresponding to patient specimen were obtained under UCLA IRB 18-001029 from the UCLA Health private database for the current study and therefore cannot be made publicly available.

## Code availability

The stain-to-stain transformation-related TensorFlow codes used in this manuscript can be found on GitHub at https://github.com/kevindehaan/stain-transformation.

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

## Acknowledgements
The authors acknowledge the funding of the NSF Biophotonics Program (USA). Mei Leng from the Department of Medicine Statistics Core at the UCLA Clinical and Translational Science Institute is also acknowledged for helping to perform the statistical analysis.

## Author contributions
T.L. imaged the unlabeled tissue sections. K.d.H. and Y.Z. processed the data. A.E.S., M.F.P.D., and W.D.W. performed the diagnoses for the initial study. A.E.S., W.D.W, and K.Y.J performed the diagnoses for the expanded study. J.E.Z. chose the cases used to test the study and performed the adjudication. A.E.S., W.D.W., and J.E.Z. performed the stain quality assessment study. A.N., S.L, S.Z., and R.R. digitized and performed quality checks on the digital slides. K.d.H., Y.Z., Y.R., W.D.W., and A.O. prepared the manuscript, and all authors contributed to the manuscript. A.O. supervised the research.

## Competing interests
Y.R. and A.O. are co-inventors of a pending patent application US20210043331A1, which covers the use of label-free autofluorescence images to generate virtually stained images. K.d.H., Y.Z., Y.R., and A.O. have a pending patent application (PCT/US2020/066708), which covers the use of the stain transformation network and the use of multiple stains being performed through a single neural network. K.d.H., Y.R., W.D.W., and A.O. have a financial interest in the commercialization of deep learning-based tissue staining. J.E.Z. is a paid consultant for Leica Biosystems. The remaining authors declare no competing interests.
