## [Peer Review File · Nature Communications]

REVIEWER COMMENTS

Reviewer #1 (Remarks to the Author):

This paper proposes a method for virtual staining of H&E-stained kidney tissue into special stain images. By generating training data using a deep neural network generating H&E and special stain images from autofluorescence images, they have created a deep neural network to transform special stain images from H&E images. The authors claim that this method is more accurate than other methods using unpaired images, such as CycleGAN because it uses precisely aligned image pairs. They also show that the virtually stained images improve diagnosis accuracy for various kidney diseases compared to H&E images alone.

The method is based on an interesting unique idea and has a high potential clinical utility. However, some of the evaluations are inadequate, and therefore the clinical utility cannot be fully proved.

Major

- The proposed method relies heavily on the performance of the virtual stain network. Virtual staining of autofluorescence images into H&E images has been evaluated in another paper, but the virtual staining into three special stain images has not been assessed. Although using all of the three virtually stained images improves diagnostic accuracy, the performance of any of the special staining may be poor. Therefore, the performance of each generator should be evaluated. For example, how about evaluating the quality of virtually stained images for which the authors have a ground truth and not used in the training? The authors could also present the virtually stained images and histochemically stained serial sections to the pathologist and ask him to compare the two images.
- In the clinical practice, it is unlikely that a pathologist will make a diagnosis based on H&E staining alone unless the pathologist determines that H&E alone is sufficient to make a diagnosis. Therefore, it is not sufficient to evaluate whether the addition of special virtual staining to H&E improves diagnostic accuracy. It is necessary to assess how the accuracy of the diagnosis changes with virtual stain and real stain of serial sections.
- Likewise, unless this stain transformation technique is perfect, it is unlikely that the actual clinical practice will end up with only virtual stains. If a special stain is performed consequently, the clinical utility is limited except for diseases that require rapid diagnosis such as ACR. However, if it is possible to determine whether the tissues should be subjected to actual special staining, some staining can be omitted and has a significant impact on clinical practice. Is it possible to evaluate this?
- In Fig. 4, it is stated that the immune cells could not be easily identified in H&E images but were identified easily in special stain images, but I do not have that impression. I think it would be better to point more specific region to show that.
- The authors argue that the proposed method is more accurate than unsupervised methods such as CycleGAN because it uses precisely aligned image pairs. This idea is very reasonable, but it would be better to show if this is true experimentally.

Minor

- Names of the neural network should be consistent. (virtual stain generator network => virtual stain network. stain transformation generator network => stain transformation network).
- Explanation about Jone's silver stain is missing.
- Comma in the last term of Equation (1) should be removed.
- It is better to post all the reports, not just the summaries in Table 1

Reviewer #2 (Remarks to the Author):

In the manuscript "Deep learning-based transformation of the H&E stain into special stains" Haan and colleagues set out to generate computer generated specialist stains from the routine H&E WSI through deep learning and cycleGANs. These special stains improved upon, and resulted in more concordance, of pathologist diagnoses when compared to the original H&E alone. This can save time, cost and tissue used in the diagnostic workflow. However, the number of patients used to test the improvement to diagnosis is very small $n=16$, and of those 16 there are variable

diagnoses of multiple diseases, the majority of which are only represented once or twice. The number of real stained tissue used as ground truth for the virtual stained network seems to not be included and the ground truth for the stain transformation network is very low. Additionally there is a lack of information about how many patient samples are used to train or used as unlabelled tissue input, also we are unaware as to what these patient samples were diagnosed with. This manuscript would need to include a far larger patient cohort to test their claims on to be acceptable for publication in this journal.

Major points:

Sample numbers:

Although the training data for the stain transformation network is augmented via virtual stained unlabelled tissue, we are not told enough about the numbers of samples to make the experiments reproducible.

How many unlabelled patient samples are used to generate virtual stains and with what diagnoses (are the representative of the diagnoses used for pathologist testing)? Is this the same number of ground truth samples?

Why are only such a small number of samples with real stains used in the initial training of the GANs? I believe 10 unique tissue section to train and 3 to validate? These sample numbers are very low to generate representative images across heterogeneous disease types and to produce reproducible algorithms. What were the diagnoses of these patients and are they representative of the pathologist diagnostic test set? How many virtual samples are used to augment this training set? What numbers of further generated data samples were utilised to augment the training data set when eg rotating and flipping? How big a training data set was there in total (real samples, virtual, and manipulated images)?

How were the 16 samples to test the pathologists selected? Why only 16? This number is already very low to test, but when taking into consideration the small number of separate diagnoses that the 16 are compiled of, $n=1$ or $n=2$ for most, it gives the study low power.

I understand the value and need for augmented and GAN generated training sets, but with collaborations in hospital lab medicine and pathology departments, from multiple institutes, one would imagine there would be a wealth of archived data (H&E and special stains) to add to the training data to make the algorithm more robust and add "real life" variability across heterogeneous diseases when generating virtual stains. Also to increase the numbers to test more thoroughly through the pathologists.

Furthermore, the increase in the real stained samples would add to the variation in staining that the authors attempt to produce using virtually generated images thereby adding much value to the workflow's application to samples over and above the small number tested here.

This paper would need to increase sample numbers throughout the entire project to make it more impactful and to draw stronger conclusions on the workflow's usefulness to aid diagnoses.

I presume upon initial diagnoses of these patients, in the actual clinical setting, the pathologists utilised the specialist stains generated in the wet-laboratory. A more accurate comparison of the automated virtual staining would be to compare actual stained H&E alongside actual stained special stains versus real H&E and virtual special stains? Without a direct comparison of current clinical workflow how do we know if this improves a pathologist's diagnosis? The comparison in this test is therefore not a fair comparison and should be corrected.

Results would be more convincing if more patients of similar disease are used. Pathologist 3 had only 2 more improvements than concordances, with 2 discordances. Pathologist 2 had 11 concordance and only 4 improvements with 1 discordance. Is this seen as clinically acceptable increase over initial diagnoses? Do these findings prove that the special stains generated are worth the disruption to the workflow that would be needed to incorporate such an AI based network into the diagnostic pipeline?

On such a small set of patients could it simply be artefact that shows up and happens to help the pathologist make a potential diagnosis, is that "improved" diagnosis actually true? Were the improved diagnoses checked with patient follow up to determine if they were real?

Eg "Another example is case #10 where two pathologists were able to provide a diagnosis of membranous nephropathy only after review of the virtually generated JMS stain." Is this real or a fluke by artefact? Larger numbers would allow this to be tested.

Minor points:

Although the points made in the introduction are interesting, the level of detail is more akin to a

review which in turn makes for an unnecessarily long introduction. I would suggest making it more succinct by covering only the relevant points for the actual research without as much detail.

The last paragraph of the introduction is more fitting for the discussion section.

The 1st sentence is rather long and cumbersome making it difficult to understand, suggest re-writing it, not a good start for the reader.

The authors state in the introduction "The standard H&E stain is relatively easy to perform and is standardized across the industry"

I would not say that H&E is standardized across the industry, many staining protocols exist that can provide a lot of variation. As the authors themselves state when trying to introduce variability into the pipeline using augmented virtual data sets.

I wouldn't recommend the use of italics in the introduction to labour a point – but this is of course up to the journals formatters.

Diagnoses in table not standardized nomenclature. Eg Crescentic GN and Crescentic glomerulonephritis. Confusing to reader and seems to further dilute the number of specific cases being tested.

How was the >3 weeks wash-out determined? Were the pathologists still influenced by the original H&E?

As detailed below, we have revised our manuscript in response to the referees' comments. The original referee comments are shown in black color, whereas for ease of communication, our answers are provided in blue. The revisions that we made to the manuscript following the reviewers' comments are highlighted in yellow.

A brief summary of the changes that have been made to the manuscript are as follows:

- The number of unique cases used to test our stain transformation technique has been increased from 16 to 58. These results are presented in **Figure 4**.
 - This expanded study found that using stain-transformed special stains results in a statistically significant improvement ($p=0.0095$) over diagnoses performed using H&E only.
- An additional study was performed with the same 58 unique cases, where the pathologists were given histochemically stained special stains alongside the H&E stained tissue. As above, these diagnoses were compared against diagnoses made with H&E only and can be seen in **Figure 4**.
 - A statistically significant improvement ($p=0.0003$) was found when using H&E and histochemically stained special stains over the use of H&E only.
- No statistically significant difference between the number of improvements, concordances and discordances was found for any of the three pathologists between the two comparisons listed above, confirming that our stain-transformed special stains perform very similar to their histochemically stained counterparts.
- The raw diagnoses and full adjudication of the above listed studies have been added as **Supplementary Data**.
- We added another new study comparing the quality of stains generated by our stain-to-stain transformation networks and histochemical staining. As shown in **Supplementary Table S1**, no statistical difference was found between the quality of the two stains (stain-transformed vs. histochemical).
- A direct comparison showing the superiority of our stain-to-stain transformation technique over CycleGANs is presented in **Supplementary Figure S3**.
- Additional information describing the original diagnoses and number of fields-of-view from each of the samples used to train and validate the different neural networks has been added to the methods section as well as **Supplementary Table S2** and **Supplementary Table S3**.

Below we also provide a list of the new or significantly changed figures and supplementary files which we have included in our revised manuscript:

- **Figure 3:** The third phase of our study – performing diagnoses using H&E and histochemically stained special stains has been added to the figure.
- **Figure 4:** A visualization of the improvements, concordances, and discordances for each case.
- **Figure 5:** This figure has been updated to allow the reader to more clearly see how the computational staining helps highlight specific cells.
- **Supplementary Figure S3:** A comparison between the performance of the stain-to-stain transformation technique presented by our paper against a CycleGAN.
- **Supplementary Table S1:** Results of the stain quality comparison study.
- **Supplementary Table S2:** A list of the original diagnoses for the samples used to train and validate the stain transformation network.
- **Supplementary Table S3:** A list of the original diagnoses and dataset sizes for the samples used to train the stain style augmentation networks.
- **Supplementary Data:** A full listing of the adjudicated results for the two diagnostic studies, alongside the raw diagnostic information input by each of the pathologists.
- **Supplementary Images:** The images given to the pathologists for the quality assessment study.

Note that the original Figure 4 has been removed (the data have been added to the Supplementary Data file), and the original Supplementary Figure S3 is now Supplementary Figure S4.

Reviewer #1 (Remarks to the Author):

This paper proposes a method for virtual staining of H&E-stained kidney tissue into special stain images. By generating training data using a deep neural network generating H&E and special stain images from autofluorescence images, they have created a deep neural network to transform special stain images from H&E images. The authors claim that this method is more accurate than other methods using unpaired images, such as CycleGAN because it uses precisely aligned image pairs. They also show that the virtually stained images improve diagnosis accuracy for various kidney diseases compared to H&E images alone.

The method is based on an interesting unique idea and has a high potential clinical utility.

However, some of the evaluations are inadequate, and therefore the clinical utility cannot be fully proved.

Major

1) The proposed method relies heavily on the performance of the virtual stain network. Virtual staining of autofluorescence images into H&E images has been evaluated in another paper, but the virtual staining into three special stain images has not been assessed. Although using all of the three virtually stained images improves diagnostic accuracy, the performance of any of the special staining may be poor. Therefore, the performance of each generator should be evaluated. For example, how about evaluating the quality of virtually stained images for which the authors have a ground truth and not used in the training? The authors could also present the virtually stained images and histochemically stained serial sections to the pathologist and ask him to compare the two images.

We thank the reviewer for their feedback. As discussed by the reviewer, virtual staining of autofluorescence images into both H&E and special stains has been evaluated in other papers (Ref 7,8 in the manuscript). In order to show that the quality of the networks performing stain transformation between H&E images and special stains is indeed sufficient, we have performed a new study where the pathologists compared the quality of our stains generated using the stain transformation network and the images of the histochemically stained tissue samples.

The following has been added to the Results section of the revised manuscript:

“...An additional study was performed to assess the quality of the stains generated by the stain transformation network. For this study, three pathologists rated the quality of various aspects of the stains generated using the stain transformation network as well as the images of histochemically stained tissue from serial tissue sections. The pathologists each viewed 16 unique rectangular FOVs (with dimensions ranging from ~150 μm \times 175 μm to ~375 μm \times 500 μm) coming from the three validation slides used during the training of our neural network. These same FOVs were scored for each of the 3 generated special stains, as well as for the same region of the tissue in a serial histochemically stained tissue section. The 16 FOVs were randomly chosen by one of the pathologists to be representative of the tissue sections used, and only in-focus areas where the tissue is unbroken in all of the tissue sections were selected. The FOVs were ordered randomly, and each pathologist rated every image twice – before and after the image randomly being rotated or flipped (resulting in a total of 32 ratings for each stain and image type). Overall, each one of the three pathologists rated 192 FOVs (half virtually stained and half histologically stained).

The pathologists scored four aspects of each FOV on a scale from 1 to 4, where 4 is perfect, 3 is very good, 2 is good enough (passable), and 1 is not acceptable, for a total of 2304 unique assessments/ratings made by three pathologists. The Masson’s trichrome stain was rated for overall stain quality, nuclear detail, cytoplasmic detail, and extracellular fibrosis quality. The PAS and Jones Silver stains were rated based on their overall stain quality, along with nuclear detail, cytoplasmic detail and basement membrane detail.

Supplementary Table S1 shows the mean score for each stain type and quality metric. This table shows that the difference in stain quality for all of the measured aspects of each stain is significantly smaller than the standard error between the ratings. This indicates that the stain-to-stain transformation technique achieves quality of stain equivalent to that of the histochemically stained

tissue used as our ground truth. A non-randomized version of the images used for this stain quality evaluation study can be found in the Supplementary Images file...”

(New) Supplementary Table S1. Quality comparison between the stains generated by the stain transformation network and the histochemically stained tissue, where 4 is Perfect, 3 is Very Good, 2 is Good Enough (Passable), and 1 is Not Acceptable. The scores are the average of 16 fields-of-view coming from 3 tissue sections, each rated twice by three separate pathologists.

a) Masson's Trichrome				
	Stain quality score	Nuclear detail	Cytoplasmic detail	Extracellular Fibrosis
Stain transformation	3.19	3.39	3.24	3.11
Histologically stained	3.09	2.95	3.19	3.30
Stain transformation std. error (between pathologists)	0.52	0.35	0.47	0.71
Std. error histological (between pathologists)	0.21	0.27	0.25	0.43
b) PAS				
	Stain quality score	Nuclear detail	Cytoplasmic detail	Basement membrane detail
Stain transformation	3.40	3.53	3.38	3.39
Histologically stained	3.51	3.49	3.41	3.53
Stain transformation std. error (between pathologists)	0.41	0.26	0.39	0.44
Std. error histological (between pathologists)	0.33	0.33	0.42	0.33
c) Jones Silver Stain				
	Stain quality score	Nuclear detail	Cytoplasmic detail	Basement membrane detail
Stain transformation	3.84	3.70	3.70	3.91
Histologically stained	3.88	3.72	3.82	3.98
Stain transformation std. error (between pathologists)	0.13	0.22	0.15	0.05
Std. error histological (between pathologists)	0.06	0.16	0.01	0.02

2) In the clinical practice, it is unlikely that a pathologist will make a diagnosis based on H&E staining alone unless the pathologist determines that H&E alone is sufficient to make a diagnosis. Therefore, it is not sufficient to evaluate whether the addition of special virtual staining to H&E improves diagnostic accuracy. It is necessary to assess how the accuracy of the diagnosis changes with virtual stain and real stain of serial sections.

We thank the reviewer for their feedback. We have added an additional study, in which the pathologists each were given WSIs of the H&E tissue section **alongside the histochemically stained special stains from adjacent tissue sections**. This study has allowed us to compare the diagnostic performance of special stains generated using our stain transformation method against those created through the standard histochemical staining. However, it is important to note that the special stains (histochemical) come from adjacent tissue sections, whereas our stain-transformed images come from a single section. As was the case with the previous studies, the diagnoses were

performed after an additional >3-week washout period. We have summarized these analyses in our revised **Results** section as follows:

“... To validate the presented stain transformation technique, a study was performed using WSI data from 58 different H&E stained tissue sections (each corresponding to a unique patient) obtained from an existing database of non-neoplastic kidney diseases. In this blinded study, three board-certified pathologists filled out diagnostic information for each H&E WSI (see the Methods section for details). Following a >3-week washout period, the same pathologists were asked to fill out the same diagnostic information, but along with the H&E, they were also provided the stain-transformed WSIs corresponding to special stains PAS, MT, and JMS, all generated from the existing H&E images. Following a second >3-week washout period, the pathologists were asked to fill out the same diagnostic information. For this third phase, instead of using computationally-generated, stain-transformed special stains, histochemically stained serial tissue sections were given to the pathologists along with the H&E (these sections originated from different depths within the tissue block). A diagram visualizing this study process can be seen in Figure 3. Following the third round of diagnoses, a fourth board-certified pathologist adjudicated all the results/diagnoses and determined whether the viewing of the neural network generated special stains resulted in an Improvement (I), Concordance (C) or Discordance (D) with respect to the original H&E-only diagnoses...”

“...Adjudication of the preliminary diagnoses made by using H&E only and the use of both H&E and stain-transformed special stains across the 58 cases revealed that using stain-to-stain transformations resulted in an average of 13 improved diagnoses (22.4%), 38.3 concordant diagnoses (66.1%) and 6.7 discordant diagnoses (11.5%) across the three pathologists. A total of 10 cases had an improved preliminary diagnosis by 2 or more pathologists, while 3 cases had a discordant diagnosis by more than one pathologists (see Figure 4). When comparing the diagnoses made with only H&E against those made with H&E alongside the histochemically stained special stains from serial tissue sections, an average of 15 improved diagnoses (25.8%), 38.6 concordant diagnoses (66.6%) and 4.3 discordant diagnoses (7.4%) were found across the three pathologists and 58 cases. For this second comparison, 12 cases were improved by two or more pathologists, while 2 cases were discordant for more than one pathologist (see Figure 4).

These results show that the additional 3 virtual special stains improve the diagnostic outcome over a single histochemically stained H&E slide for a myriad of non-neoplastic diseases ($P=0.0095$, using a one-tailed t -test). Our stain-to-stain transformation results are also in line with the level of improvement demonstrated when the pathologists had access to the H&E and 3 additional sections that are histochemically stained with the corresponding special stains ($P=0.0003$, using a one-tailed t -test) over a single histochemically stained H&E slide. In addition to these, a secondary analysis was used to compare differences in the proportion of improvements, concordances and discordances for each of these two comparisons reported in Figure 4a,b. To do this, three separate chi-square tests were used – one for each pathologist (see the Methods section). These tests found that, while the histochemically stained tissue performed better for all three pathologists, the differences between the two comparisons were not statistically significant (with P values of 0.60, 0.34, and 0.92 for the first, second, and third pathologist, respectively).”

Further details of the Statistical Analysis provided above have been added to the revised **Methods** section:

“...Using the preliminary study of 16 samples, we calculated that a total of 41 samples are needed to show statistical significance (using a power of 0.8 and an alpha level of 0.05 and using a one tailed t -test). Therefore, the total number of patients was increased 58 to ensure that the study was sufficiently powered.

A one tailed t -test was used to determine whether a statistically significant number of improvements were made when using either [H&E and stain-transformed special stains], or [H&E and histochemically stained special stains] over only [H&E] images. The statistical analyses were performed by giving a score of +1 to any improvement, -1 to any discordance and 0 to any

concordance. The score for each case was then averaged among the three pathologists who evaluated the case, and the test showed that the amount of improvement (i.e. if the average score is greater than zero) across the 58 cases was statistically significant.

A chi-squared test with two degrees of freedom was used to compare the proportion of improvements, concordances and discordances between the methods tested above. The improvements, concordances and discordances for each pathologist was compared individually.

For all tests, a P value of 0.05 or less was considered to be significant.”

3) Likewise, unless this stain transformation technique is perfect, it is unlikely that the actual clinical practice will end up with only virtual stains. If a special stain is performed consequently, the clinical utility is limited except for diseases that require rapid diagnosis such as ACR. However, if it is possible to determine whether the tissues should be subjected to actual special staining, some staining can be omitted and has a significant impact on clinical practice. Is it possible to evaluate this?

Thank you for these thoughtful comments. We agree, in the setting of renal pathology, the H&E stain is very rarely sufficient for the final diagnosis. However, in most practices, the H&E stains become available before the special stains and preliminary diagnoses are sometime rendered when there is diagnostic urgency (e.g. concern for rapidly progressive glomerulonephritis or rejection in the transplant setting, etc.). There is often an overnight delay between the availability of the H&E stains and the special stains. Therefore, we do feel there is merit in evaluating if the virtual special stains add diagnostic value to the H&E alone, especially in the setting of a need for an urgent renal pathologic diagnosis.

We have clarified this point in our revised manuscript as:

“...Non-neoplastic kidney disease relies on special stains to provide the standard of care pathologic evaluation. In many clinical practices, H&E stains are available well before the special stains are prepared, and pathologists may provide a preliminary diagnosis to enable the patient’s nephrologist to begin any necessary treatment. In a setting when only H&E slides are initially available, the preliminary diagnosis is followed by the final diagnosis made by examining the special stain images, which are often provided the next working day. Using the presented stain transformation technique (Figure 1) would alleviate the need to wait for the special stains to be available. This is especially useful for some urgent medical conditions such as crescentic glomerulonephritis or transplant rejection where quick and accurate diagnosis followed by rapid initiation of treatment may lead to significant improvements in clinical outcomes.”

There are a few aspects to consider when comparing virtual stains with chemically stained slides on serial sections. First of all, similar to the validation of digital pathology systems in general, perfection is not required, and the standard of practice is to demonstrate lack of inferiority - which is what we have endeavored to do. Substituting chemically stained slides for virtually stained slides has several advantages including decreased slide preparation time, decreased laboratory costs and preservation of tissue for subsequent analysis, if necessary. As the reviewer pointed to, review of the virtual stain may even make it possible to selectively omit the need for performing actual special stains and save time and laboratory expenses in some settings. However, it is beyond the scope of this study to objectively assess this potential change to laboratory practice.

Following the referee’s comment, we have added the following to the Discussion section of the revised manuscript:

“Similar to the validation of digital pathology systems in general, a perfect stain-to-stain transformation is not required; the standard of practice is to demonstrate lack of inferiority, which is what we have endeavored to do. Substituting chemically stained slides for stain-transformed slides leads to several advantages, including e.g., decreased slide preparation time, decreased laboratory costs and preservation of tissue for subsequent analysis, if necessary. In the future, the use of computationally generated special stains may make it possible to selectively omit the need for performing actual special stains and save time and laboratory expenses in some settings.”

4) In Fig. 4, it is stated that the immune cells could not be easily identified in H&E images but were identified easily in special stain images, but I do not have that impression. I think it would be better to point more specific region to show that.

Thank you for highlighting this point. We intended to point out that the location of the immune cells (specifically inside or outside of tubules) is difficult to determine on a conventional H&E stain. The introduction of the basement membrane staining with the virtual PAS and JMS stains certainly helps with determining the location of the cellular infiltrate. We have modified the manuscript to state:

“This biopsy contains very pronounced cellular inflammation that is difficult to precisely localize on a standard H&E stain, as H&E does not give clear contrast to structures such as basement membranes. The computationally generated special stains highlight the tubular basement membranes which allowed all 3 pathologists to see the location of the inflammatory cells and give a more precise characterization of the organ rejection process.”

We have also **modified Figure 5** and its captions accordingly to show the exact location, as can be seen below:

Figure 5: Examples of improved diagnoses fostered by the stain-transformed special stains. We report here WSIs that are generated using the stain transformation technique. In this case, the addition of the computationally generated special stains improved all three of the diagnoses made by the pathologists. The red arrows point to a region, where the special stains help highlight inflammatory cells within the tubule, otherwise the boundary of the tubules cannot be seen with the H&E stain only. (i) Generation of JMS. (ii) Generation of MT. (iii) Generation of PAS.

5) The authors argue that the proposed method is more accurate than unsupervised methods such as CycleGAN because it uses precisely aligned image pairs. This idea is very reasonable, but it would be better to show if this is true experimentally.

Following the referee's suggestion, we have added the following explanation and new analyses on this topic to the **Discussion** section of the revised manuscript:

*“...When the same CycleGAN architecture used for the data augmentation is applied to the various stain transformations, a number of clear hallucinations occur. These hallucinations are particularly evident for the PAS and Jones Silver stain, where the networks incorrectly label the tubular basement membranes (see **Supplementary Figure S3**). The tubules are composed of epithelial cells lining basement membranes that stain black on the Jones stain and magenta on the PAS stain. The brush*

border lining the luminal surface of the epithelial cells is also normally lightly stained black and magenta by the Jones and PAS stains, respectively. The CycleGAN method incorrectly recognized the basement membranes and tubular brush borders leading to incorrect image generation, which is a very significant error. In contrast, the quality and features of the Masson's Trichrome stain appear to be more similar between the two techniques. This is believed to be due to the Masson's Trichrome stain being relatively similar to H&E, while the other stains require significant structural changes which can cause the hallucinations for CycleGANs. These results and observations highlight the significant advantages of our stain-to-stain transformation network compared to standard CycleGAN-based methods."

Supplementary Figure S3: Comparison between the performance of the stain-transformation

network presented in this work and stain transformations performed by a CycleGAN. For the Masson's Trichome stain there are only minor differences between the quality of the CycleGAN and our method. However, the other two stains are much more difficult for a CycleGAN to perform, and the CycleGAN technique hallucinates features throughout all of the images generated for these stains. For example, the red arrows point to locations where the basement membrane has been incorrectly labeled by the CycleGAN for the Jones and PAS stains.

Minor

6) Names of the neural network should be consistent. (virtual stain generator network => virtual stain network. stain transformation generator network => stain transformation network).

In an effort to make the distinction between the various networks used in the manuscript, we have made modifications to ensure that the “*virtual staining network*” refers to the virtual staining of autofluorescence images, and that the “*stain transformation*” network refers to the transformation from H&E to special stains.

As discussed in the manuscript, a Generative Adversarial Network (GAN) is used during the training phase, both the “virtual stain network” and the “stain transformation network” use two distinct networks (a “generator” network and a “discriminator” network). Therefore, when we discuss the “generator networks” we are specifically referring to that portion of the neural network rather than to either the virtual staining network or stain transformation network.

7) Explanation about Jones's silver stain is missing.

We thank the reviewer for their feedback and have accordingly added the following explanation about Jones's silver stain to our revised manuscript:

“The black staining in the Jones Methenamine silver stain offers sharp contrast to visualize glomerular architecture and enables the pathologist to recognize subtle basement membrane abnormalities resulting from remodeling due to various forms of injury.”

8) Comma in the last term of Equation (1) should be removed.

We have removed the comma from Equation (1).

9) It is better to post all the reports, not just the summaries in Table 1

We have added the full reports as Supplementary Data.

Reviewer #2 (Remarks to the Author):

In the manuscript “Deep learning-based transformation of the H&E stain into special stains” Haan and colleagues set out to generate computer generated specialist stains from the routine H&E WSI through deep learning and cycleGANs. These special stains improved upon, and resulted in more concordance, of pathologist diagnoses when compared to the original H&E alone. This can save time, cost and tissue used in the diagnostic workflow. However, the number of patients used to test the improvement to diagnosis is very small $n=16$, and of those 16 there are variable diagnoses of multiple diseases, the majority of which are only represented once or twice. The number of real stained tissue used as ground truth for the virtual stained network seems to not be included and the ground truth for the stain transformation network is very low. Additionally there is a lack of information about how many patient samples are used to train or used as unlabelled tissue input, also we are unaware as to what these patient samples were diagnosed with. This manuscript would need to include a far larger patient cohort to test their claims on to be acceptable for publication in this journal.

Major points:

Sample numbers:

1) Although the training data for the stain transformation network is augmented via virtual stained unlabelled tissue, we are not told enough about the numbers of samples to make the experiments reproducible.

The Methods section of the manuscript details the number of samples and patches used to train the

stain transformation neural networks, and revisions have been made so that it is more clear how many samples and patches were used. The statement now reads as follows:

“...The stain transformation networks were trained using pairs of 256x256-pixel image patches generated by the class conditional virtual staining network (label-free), downsampled by a factor of 2 (to match 20x magnification). These patches were randomly cropped from one of 1013 712x712-pixel images coming from 10 unique tissue sections, leading to ~7,836 unique patches usable for training. 76 additional images coming from three unique tissue sections were used to validate the network. These images were augmented using the eight stain augmentation networks, and further augmented through random rotation and flipping of the images.”

To ensure that there is enough information to make the augmentation networks reproducible, we have added a table listing the number of patches used to train each of the CycleGAN stain augmentation networks to the supplementary (see **Table S3**).

As will be discussed below, we have more thoroughly detailed the number of images and the number of samples used to train the virtual staining, and stain transformation neural networks. We have also added the diagnoses of the samples used in the training and validation data to the **Supplementary Information, Tables S2 and S3**.

2) How many unlabelled patient samples are used to generate virtual stains and with what diagnoses (are the representative of the diagnoses used for pathologist testing)? Is this the same number of ground truth samples?

The virtual staining networks were trained using co-registered pairs of images coming from the same tissue section before and after staining. Therefore, the unlabeled patient tissue sections are the same tissue as the ground truth samples. The only networks which do not rely on matched data (i.e. where the input image comes from a different tissue section than the ground truth) are the CycleGAN networks used to perform only the input data augmentation.

The datasets used to train the virtual staining networks for the four different stains (H&E, PAS, Masson's Trichrome and Jones Silver stain) all used adjacent tissue sections from the same patients. We have added the following information about the samples, and the number of patches used to train the virtual staining networks:

*“...A set of four adjacent tissue sections were used to train the virtual staining networks for H&E and the three special stains. The H&E portion of all three of the networks was trained with 1058 1424x1424-pixel images coming from 10 unique patients, the PAS network was trained with 946 1424x1424-pixel images coming from 11 unique patients, the Jones network was trained with 816 1424x1424-pixel images coming from 10 unique patients, and the Masson's Trichrome network was trained with 966 1424x1424-pixel images coming from 10 unique patients. A list of the samples used to train the various networks, and the original diagnoses of the patients can be seen in **Supplementary Table S2**. All of the stains were validated using the same three validation slides.”*

We have clarified that the unlabeled tissue and the ground truth samples are the same tissue sections before and after staining. As discussed above, we have also added the diagnoses of the samples used in the training and validation data to the Supplementary Information.

3) Why are only such a small number of samples with real stains used in the initial training of the GANs? I believe 10 unique tissue section to train and 3 to validate? These sample numbers are very low to generate representative images across heterogeneous disease types and to produce reproducible algorithms. What were the diagnoses of these patients and are they representative of the pathologist diagnostic test set? How many virtual samples are used to augment this training set? What numbers of further generated data samples were utilised to augment the training data set when eg rotating and flipping? How big a training data set was there in total (real samples, virtual, and manipulated images)?

The dataset used to perform the training is sufficiently large. As discussed above and written in the manuscript, **Methods** section:

“...The stain transformation networks were trained using pairs of 256x256-pixel image patches

generated by the class conditional virtual staining network (label-free), downsampled by a factor of 2 (to match 20x magnification). These patches were randomly cropped from one of 1013 712x712-pixel images coming from 10 unique tissue sections, leading to ~7,836 unique patches usable for training. 76 additional images coming from three unique tissue sections were used to validate the network. These images were augmented using the eight stain augmentation networks, and further augmented through random rotation and flipping of the images.”

Our revised **Methods** section further emphasizes this point:

“... As this stain-transformation neural network performs an image-to-image transformation, it learns to transform specific structures using the ~513 million pixels in the dataset that are independently accounted for in the loss function. Furthermore, since the network learns to convert structures which are common throughout many different types of samples, it can be applied to tissues with diseases that the network was not trained with. When used in conjunction with the 8 data augmentation networks which convert the values of these pixels, as well as random rotation and flipping (for an additional 8x) augmentation, **there are effectively many billions of pixels which are used to learn the desired stain-to-stain transformation. Because of these advantages, a much smaller number of training samples from unique patients can be used than would be required for a typical classification neural network...**”

Therefore, the datasets used to train our neural networks are sufficient in size. If the purpose of our neural network instead was to automatically perform a diagnosis, several orders of magnitude more data would be needed.

In our revised manuscript, we have also performed an additional study showing that the quality of the special stains generated by our stain transformation networks is equivalent to that of histochemically stained tissue, which further reinforces the claim that our stain transformation networks are effective with the current training dataset. Details of this study are explained in the answer to the first question by Reviewer 1 (also see the new **Supplementary Table S1**).

Finally, as discussed above, we have also added the diagnoses of the patients to the Supplementary Information file.

4) How were the 16 samples to test the pathologists selected? Why only 16? This number is already very low to test, but when taking into consideration the small number of separate diagnoses that the 16 are compiled of, n=1 or n=2 for most, it gives the study low power.

We thank the reviewer for their input. The samples to test our approach were chosen to cover a wide variety of non-neoplastic kidney diseases. Furthermore, we only chose samples where we were able to access whole slide images of the four histological stains coming from the same tissue block.

In regards to the number of samples - we have significantly increased the number of samples blindly tested to a total of 58 in order to increase the power of the study.

We have summarized these new analyses in our revised our **Results** section as follows:

“... To validate the presented stain transformation technique, a study was performed using WSI data from 58 different H&E stained tissue sections (each corresponding to a unique patient) obtained from an existing database of non-neoplastic kidney diseases. In this blinded study, three board-certified pathologists filled out diagnostic information for each H&E WSI (see the Methods section for details). Following a >3-week washout period, the same pathologists were asked to fill out the same diagnostic information, but along with the H&E, they were also provided the stain-transformed WSIs corresponding to special stains PAS, MT, and JMS, all generated from the existing H&E images. Following a second >3-week washout period, the pathologists were asked to fill out the same diagnostic information. For this third phase, instead of using computationally-generated, stain-transformed special stains, histochemically stained serial tissue sections were given to the pathologists along with the H&E (these sections originated from different depths within the tissue

block). A diagram visualizing this study process can be seen in Figure 3. Following the third round of diagnoses, a fourth board-certified pathologist adjudicated all the results/diagnoses and determined whether the viewing of the neural network generated special stains resulted in an Improvement (I), Concordance (C) or Discordance (D) with respect to the original H&E-only diagnoses...

“...Adjudication of the preliminary diagnoses made by using H&E only and the use of both H&E and stain-transformed special stains across the 58 cases revealed that using stain-to-stain transformations resulted in an average of 13 improved diagnoses (22.4%), 38.3 concordant diagnoses (66.1%) and 6.7 discordant diagnoses (11.5%) across the three pathologists. A total of 10 cases had an improved preliminary diagnosis by 2 or more pathologists, while 3 cases had a discordant diagnosis by more than one pathologists (see Figure 4). When comparing the diagnoses made with only H&E against those made with H&E alongside the histochemically stained special stains from serial tissue sections, an average of 15 improved diagnoses (25.8%), 38.6 concordant diagnoses (66.6%) and 4.3 discordant diagnoses (7.4%) were found across the three pathologists and 58 cases. For this second comparison, 12 cases were improved by two or more pathologists, while 2 cases were discordant for more than one pathologist (see Figure 4).

These results show that the additional 3 virtual special stains improve the diagnostic outcome over a single histochemically stained H&E slide for a myriad of non-neoplastic diseases ($P=0.0095$, using a one-tailed t -test). Our stain-to-stain transformation results are also in line with the level of improvement demonstrated when the pathologists had access to the H&E and 3 additional sections that are histochemically stained with the corresponding special stains ($P=0.0003$, using a one-tailed t -test) over a single histochemically stained H&E slide. In addition to these, a secondary analysis was used to compare differences in the proportion of improvements, concordances and discordances for each of these two comparisons reported in Figure 4a,b. To do this, three separate chi-square tests were used – one for each pathologist (see the Methods section). These tests found that, while the histochemically stained tissue performed better for all three pathologists, the differences between the two comparisons were not statistically significant (with P values of 0.60, 0.34, and 0.92 for the first, second, and third pathologist, respectively).”

Further details of the **Statistical Analysis** provided above have been added to the revised **Methods** section:

“...Using the preliminary study of 16 samples, we calculated that a total of 41 samples are needed to show statistical significance (using a power of 0.8 and an alpha level of 0.05 and using a one tailed t -test). Therefore, the total number of patients was increased 58 to ensure that the study was sufficiently powered.

A one tailed t -test was used to determine whether a statistically significant number of improvements were made when using either [H&E and stain-transformed special stains], or [H&E and histochemically stained special stains] over only [H&E] images. The statistical analyses were performed by giving a score of +1 to any improvement, -1 to any discordance and 0 to any concordance. The score for each case was then averaged among the three pathologists who evaluated the case, and the test showed that the amount of improvement (i.e. if the average score is greater than zero) across the 58 cases was statistically significant.

A chi-squared test with two degrees of freedom was used to compare the proportion of improvements, concordances and discordances between the methods tested above. The improvements, concordances and discordances for each pathologist was compared individually.

For all tests, a P value of 0.05 or less was considered to be significant..”

5) I understand the value and need for augmented and GAN generated training sets, but with collaborations in hospital lab medicine and pathology departments, from multiple institutes, one would imagine there would be a wealth of archived data (H&E and special stains) to add to the training data to make the algorithm more robust and add “real life” variability across heterogeneous diseases when generating virtual stains. Also to increase the numbers to test more thoroughly through the pathologists.

Furthermore, the increase in the real stained samples would add to the variation in staining that the authors attempt to produce using virtually generated images thereby adding much value to the workflow's application to samples over and above the small number tested here.

The samples to test our approach were chosen to cover a wide variety of non-neoplastic kidney diseases, and we only chose samples where we were able to access whole slide images of the four histological stains coming from the same tissue block.

To address the referee's comments above, we have added the following statement to the **Discussion** section of the revised manuscript:

"...It is important to note that the current stain-to-stain network is trained to work with H&E stains performed at a few institutions and imaged by different microscopes from the same vendor/model (Leica Biosystems Aperio AT2 slide scanner). Additional data would be required for the network to generalize to samples imaged using microscopes with different specifications or vendors, or any H&E stains which are performed in a significantly different manner. Furthermore, while this study covers a broad range of diseases, it is still a proof-of-concept. Future studies should be performed which contain both larger training and test datasets in order to conclusively show the technique may be suitable for diagnostic use. Future work may also apply this technique that we have presented to other biomarkers that are currently labeled with IHC to help target specific conditions..."

6) This paper would need to increase sample numbers throughout the entire project to make it more impactful and to draw stronger conclusions on the workflow's usefulness to aid diagnoses.

We thank the reviewer for their feedback. We have significantly increased the sample numbers used for testing and performed additional analyses to help draw stronger conclusions of the workflow's usefulness, as summarized below:

- The number of unique cases used to test our stain transformation technique has been increased from 16 to 58. These results are presented in **Figure 4**.
 - This expanded study found that using stain-transformed special stains results in a statistically significant improvement ($p=0.0095$) over diagnoses performed using H&E only.
- An additional study was performed with the same 58 unique cases, where the pathologists were given histochemically stained special stains alongside the H&E stained tissue. As above, these diagnoses were compared against diagnoses made with H&E only and can be seen in **Figure 4**.
 - A statistically significant improvement ($p=0.0003$) was found when using H&E and histochemically stained special stains over the use of H&E only.
- No statistically significant difference between the number of improvements, concordances and discordances was found for any of the three pathologists between the two comparisons listed above, confirming that our stain-transformed special stains perform very similar to their histochemically stained counterparts.
- The raw diagnoses and full adjudication of the above listed studies have been added as **Supplementary Data**.
- We added another new study comparing the quality of stains generated by our stain-to-stain transformation networks and histochemical staining. As shown in **Supplementary Table S1**, no statistical difference was found between the quality of the two stains (stain-transformed vs. histochemical).
- A direct comparison showing the superiority of our stain-to-stain transformation technique over CycleGANs is presented in **Supplementary Figure S3**.
- Additional information describing the original diagnoses and number of fields-of-view from each of the samples used to train and validate the different neural networks has been added to the methods section as well as **Supplementary Table S2** and **Supplementary Table S3**.

7) I presume upon initial diagnoses of these patients, in the actual clinical setting, the pathologists utilised the specialist stains generated in the wet-laboratory. A more accurate comparison of the automated virtual staining would be to compare actual stained H&E alongside actual stained special stains versus real H&E and virtual special stains? Without a direct comparison of current clinical workflow how do we know if this improves a pathologist's diagnosis? The comparison in this test is therefore not a fair comparison and should be corrected.

We thank the reviewer for their feedback. This question has been answered above in our responses to Questions 4 and 6.

To summarize:

- The number of unique cases used to test our stain transformation technique has been increased from 16 to 58. These results are presented in **Figure 4**.
 - This expanded study found that using stain-transformed special stains results in a statistically significant improvement ($p=0.0095$) over diagnoses performed using H&E only.
- An additional study was performed with the same 58 unique cases, where the pathologists were given histochemically stained special stains alongside the H&E stained tissue. As above, these diagnoses were compared against diagnoses made with H&E only and can be seen in **Figure 4**.
 - A statistically significant improvement ($p=0.0003$) was found when using H&E and histochemically stained special stains over the use of H&E only.
- No statistically significant difference between the number of improvements, concordances and discordances was found for any of the three pathologists between the two comparisons listed above, confirming that our stain-transformed special stains perform very similar to their histochemically stained counterparts.
- The raw diagnoses and full adjudication of the above listed studies have been added as **Supplementary Data**.
- We added another new study comparing the quality of stains generated by our stain-to-stain transformation networks and histochemical staining. As shown in **Supplementary Table S1**, no statistical difference was found between the quality of the two stains (stain-transformed vs. histochemical).

Further details of the **Statistical Analysis** provided above have been added to the revised **Methods** section (also see our response above to Question #4).

8) Results would be more convincing if more patients of similar disease are used.

As discussed above in our earlier responses, we have significantly increased the number of samples used for testing which has in turn allowed more extensive statistical analysis.

Pathologist 3 had only 2 more improvements than concordances, with 2 discordances. Pathologist 2 had 11 concordance and only 4 improvements with 1 discordance. Is this seen as clinically acceptable increase over initial diagnoses? Do these findings prove that the special stains generated are worth the disruption to the workflow that would be needed to incorporate such an AI based network into the diagnostic pipeline?

As discussed in the manuscript, we believe that one of the main applications of this technique would be for preliminary diagnoses where only histochemically stained H&E tissue would be prepared. The H&E stains become available before the special stains and preliminary diagnoses are sometime rendered when there is diagnostic urgency (e.g. concern for rapidly progressive glomerulonephritis or rejection in the transplant setting, etc.). For example, there is often an overnight delay between the availability of the H&E stains and the special stains. Therefore, we do feel there is merit in evaluating if the virtual special stains add diagnostic value to the H&E alone, especially in the setting of a need for an urgent renal pathologic diagnosis.

We have clarified these points in our revised manuscript as:

“Non-neoplastic kidney disease relies on special stains to provide the standard of care pathologic evaluation. In many clinical practices, H&E stains are available well before the special stains are prepared, and pathologists may provide a preliminary diagnosis to enable the patient’s nephrologist to begin any necessary treatment. In a setting when only H&E slides are initially available, the preliminary diagnosis is followed by the final diagnosis made by examining the special stain images, which are often provided the next working day. Using the presented stain transformation technique (Figure 1) would alleviate the need to wait for the special stains to be available. This is especially useful for some urgent medical conditions such as crescentic glomerulonephritis or transplant rejection where quick and accurate diagnosis followed by rapid initiation of treatment may lead to significant improvements in clinical outcomes.”

On such a small set of patients could it simply be artefact that shows up and happens to help the pathologist make a potential diagnosis, is that “improved” diagnosis actually true? Were the improved diagnoses checked with patient follow up to determine if they were real?

Eg “Another example is case #10 where two pathologists were able to provide a diagnosis of membranous nephropathy only after review of the virtually generated JMS stain.” Is this real or a fluke by artefact? Larger numbers would allow this to be tested.

We used the official diagnosis as our ground truth for this study. Additional information such as electron microscopy and immunofluorescence (along with the histochemically stained special stains) were also used to create these reported ground truth diagnoses. This point has also been emphasized in our revised **Results** section:

“...It is important to note that the official reported diagnosis that we used as our ground truth for this study also utilized additional information, such as electron microscopy and immunofluorescence images in order to make these diagnoses. The complete diagnostic information given by each pathologist, along with the adjudication for each report can be found in the Supplementary Data.”

Minor points:

9) Although the points made in the introduction are interesting, the level of detail is more akin to a review which in turn makes for an unnecessarily long introduction. I would suggest making it more succinct by covering only the relevant points for the actual research without as much detail.

We thank the reviewer for their feedback. The introduction has been edited to ensure that the information included is all relevant.

10) The last paragraph of the introduction is more fitting for the discussion section.

The 1st sentence is rather long and cumbersome making it difficult to understand, suggest re-writing it, not a good start for the reader.

We thank the reviewer for their feedback. We have moved this paragraph into the **Results** section of the manuscript and edited it accordingly. The paragraph now reads:

“...In order to prove the utility of our stain-transformation technique, we investigated whether it can be used to improve preliminary diagnoses made by pathologists when only H&E is available. To do this, we used stain-to-stain transformation networks to create three additional computationally-generated special stains, i.e., PAS, MT and Jones methenamine silver (JMS), from existing H&E tissue sections. These WSIs were reviewed alongside the existing histochemically-stained H&E images by pathologists (i.e., entirely bypassing the need to stain and wait for new slides). Based on tissue samples from 58 unique patients that are evaluated by three independent renal pathologists (i.e., N=174 total diagnoses), our results revealed that the generation of the three stain-transformed special stains (PAS, MT and JMS) improved the diagnoses in various non-neoplastic kidney diseases. These computationally generated panels of special stains transformed from existing H&E images using deep learning give the pathologists the additional information channels needed for

standard of patient care. We show that this unique stain-to-stain transformation workflow can be applied to a variety of diseases, and significantly improves the quality of the preliminary diagnosis when additional special stains are needed. We believe that this technique has significant utility in enhancing preliminary diagnoses, and could also provide time savings and help to reduce healthcare costs and burden for histopathology labs and patients.”

11) The authors state in the introduction “The standard H&E stain is relatively easy to perform and is standardized across the industry”

I would not say that H&E is standardized across the industry, many staining protocols exist that can provide a lot of variation. As the authors themselves state when trying to introduce variability into the pipeline using augmented virtual data sets.

We have clarified the statement as:

“The H&E stain is relatively easy to perform and is widely used across the industry”.

12) I wouldn't recommend the use of italics in the introduction to labour a point – but this is of course up the journals formatters.

We thank the reviewer for their feedback, and have edited the manuscript accordingly

13) Diagnoses in table not standardized nomenclature. Eg Crescentic GN and Crescentic glomerulonephritis. Confusing to reader and seems to further dilute the number of specific cases being tested.

We thank the reviewer for their feedback, and have accordingly standardized the nomenclature in the expanded adjudication worksheet. We note that the “raw” diagnoses input by the pathologists during the course of the three studies have been left unaltered.

14) How was the >3 weeks wash-out determined? Were the pathologists still influenced by the original H&E?

The whole slide image validation guideline produced by the College of American Pathologists Pathology and Laboratory Quality Center recommends a two-week washout period between reviewing cases for a clinical validation (see the Pantanowitz reference below). We added an additional week to our washout period for the logistical convenience of the participating pathologists. As we did not shorten the recommended washout time, we therefore do not believe that the original viewing of the H&E influenced the pathologists' interpretations. We have added an explanation indicating this in the “Pathologic evaluation of kidney biopsies” section of the revised **Methods**:
“... This >3-week washout period was chosen to be one week greater than the College of American Pathologists Pathology and Laboratory Quality Center guidelines³³, ensuring that the pathologists were not influenced by previous diagnoses.”

Pantanowitz L, et al. Validating whole slide imaging for diagnostic purposes in pathology: guideline from the College of American Pathologists Pathology and Laboratory Quality Center. Arch Pathol Lab Med. 2013 Dec;137(12):1710-22.

To conclude, we sincerely thank the referees for their constructive comments and feedback, which helped us to further improve the quality and clarity of our manuscript.

REVIEWERS' COMMENTS

Reviewer #1 (Remarks to the Author):

I am satisfied by the authors' responses to my comments and questions.